

# Plant functional diversity affects climate–vegetation interaction

Vivienne P. Groner[1,2], Thomas Raddatz[1], Christian H. Reick[1], and Martin Claussen[1,3]

[1]Max Planck Institute for Meteorology, Bundesstraße 53, 20146 Hamburg, Germany
[2]International Max Planck Research School on Earth System Modelling, Bundesstraße 53, 20146 Hamburg, Germany
[3]Center for Earth system research and Sustainability, Universität Hamburg, Bundesstraße 53, 20146 Hamburg, Germany

*Correspondence to:* V. P. Groner (vivienne.groner@mpimet.mpg.de)

**Abstract.**

We present how variations in plant functional diversity affect climate–vegetation interaction towards the end of the African Humid Period (AHP) in coupled land-atmosphere simulations using the Max Planck Institute Earth System Model MPI-ESM. In experiments with AHP boundary conditions, the extent of the "green" Sahara varies considerably with changes in plant functional diversity. Differences in vegetation cover extent and Plant Functional Type (PFT) composition translate into significantly different land surface parameters, water cycling and surface energy budget. These changes have not only regional consequences but considerably alter large scale atmospheric circulation patterns and the position of the tropical rain belt. Towards the end of the AHP, simulations with the standard PFT set in MPI-ESM depict a gradual decrease of precipitation and vegetation cover over time, while simulations with modified PFT composition show either a sharp decline of both variables or an even slower retreat. Thus, not the quantitative but the qualitative PFT composition determines climate–vegetation interaction and the climate–vegetation system response to external forcing. The sensitivity of simulated system states to changes in PFT composition raises the question how realistically Earth system models can actually represent climate–vegetation interaction, considering the poor representation of plant diversity in the current generation of land surface models.

## 1 Introduction

The "African Humid Period" (AHP) is an exceptionally interesting period to study drastic climate and vegetation changes in the past. During this period around 11,700 to 4,200 years ago, rainfall was substantially higher than today across much of West and North Africa (Bartlein et al., 2011; Shanahan et al., 2015), rivers and lakes were widespread (Hoelzmann et al., 1998; Kröpelin et al., 2008; Lézine et al., 2011a; Drake et al., 2011), and a diverse savanna-like mosaic of xeric and tropical species covered large areas of the nowadays hyperarid Sahara and arid Sahel region (Watrin et al., 2009; Hély et al., 2014). The establishment of this so-called "green" Sahara (Ritchie and Haynes, 1987; Jolly et al., 1998) was presumably triggered by changes in the Earth's orbit resulting in a stronger insolation and higher temperatures in the boreal summer than today, accompanied by an intensification and northward shifted West African monsoon (Kutzbach, 1981; Kutzbach and Guetter, 1986). It has been shown that these externally triggered changes were amplified by internal feedback mechanisms involving ocean (Kutzbach and Liu, 1997; Braconnot et al., 1999; Liu et al., 2004), surface water coverage by lakes and wetlands (Coe and Bonan, 1997; Krinner et al., 2012), dust (Pausata et al., 2016; Egerer et al., 2016), soil albedo (Knorr and Schnitzler, 2006; Vamborg et al., 2011), and




vegetation (Claussen and Gayler, 1997; Texier et al., 1997; Doherty et al., 2000; Rachmayani et al., 2015). However, despite increasing understanding of these involved feedbacks, the extent of the "green" Sahara substantially differed between previous modelling studies and all models underestimated the northward extent reconstructed from palaeo proxy data (see Claussen et al., 2017).

Similar to the establishment of the "green" Sahara, there is scientific consensus that the desertification of the southern Sahara and Sahel region towards the end of the AHP was triggered by a gradual decline in incoming solar radiation due to changes in the Earth's orbit which caused a weakening and southward shift of the West African monsoon. Conversely, the timing and abruptness of the transition from the "green" Sahara to the "desert" state are as yet uncertain, among other things because the relevance of climate–vegetation feedback in this context is still under debate. Under the assumption of a strong positive

climate–vegetation feedback, multiple stable equilibria could exist for the Sahara region – a "green" state with high vegetation cover and a "desert" state without vegetation (Claussen, 1994; Claussen and Gayler, 1997; Brovkin et al., 1998; Bathiany et al., 2012) – and the potential non-linearity of this feedback might cause an abrupt transition between these states when the system reaches a "tipping point" (Williams et al., 2011). While some studies indicated such an abrupt collapse of vegetation towards the end of the AHP implying a strong climate–vegetation feedback (Claussen et al., 1999; deMenocal et al., 2000), others

suggested a more gradual decline of vegetation (Kröpelin et al., 2008; Lézine, 2009; Lézine et al., 2011b) and precipitation (Francus et al., 2013) or attributed the collapse to other triggers (Liu et al., 2007; Rachmayani et al., 2015) thereby questioning the existence of a strong climate–vegetation feedback.

     Based on a conceptual modelling study, Claussen et al. (2013) proposed that these different viewpoints are not contradicting if one accounts for plant diversity. High plant diversity in terms of moisture requirements could stabilize a climate–vegetation

system by buffering strong feedbacks between individual plant types and precipitation, whereas a reduction in plant diversity might allow for an abrupt regime shift under gradually changing environmental conditions. Hence, climate–vegetation feedback strength would not be a universal property of a certain region but also depend on the vegetation composition. An ecological assessment of the conceptual approach by Claussen et al. (2013) and an adjustment of their model to AHP plant types corroborated their results (Groner et al., 2015). These findings raise the question how the representation of plant functional

diversity influences climate–vegetation interaction in models of higher complexity and thereby affects the extent of the "green" Sahara as well as the timing and rate of the transition to the "desert" state. To our knowledge, no previous comprehensive modelling study on the AHP has explicitly considered the effect of plant functional diversity on climate–vegetation interaction. To close this gap, we present here a series of coupled land-atmosphere simulations from mid-Holocene to preindustrial with different combinations of Plant Functional Types (PFTs) using the Max Planck Institute Earth System Model MPI-ESM. With

our idealized set up, we do not expect our simulations to match reconstructions, rather we focus on qualitative differences between simulations to find mechanisms relevant for the question how PFT diversity affects climate–vegetation interaction.





## 2   Model set up

### 2.1   MPI-ESM

We use for our study the Max Planck Institute Earth System Model MPI-ESM, a comprehensive Earth system model that couples model components for the atmosphere (ECHAM6, Stevens et al. (2013)), ocean (MPIOM, Jungclaus et al. (2013))
and land surface (JSBACH, Raddatz et al. (2007); Reick et al. (2013)) through the exchange of energy, momentum, water and carbon dioxide. This study focuses on the coupling between the atmospheric component and the land surface component.

ECHAM6 is an atmospheric General Circulation Model (GCM) which was developed at the Max Planck Institute for Meteorology in Hamburg, Germany. The model focuses on the coupling between diabatic processes and large-scale circulations which are both driven by solar insolation. For each time step the model determines the large-scale horizontal circulation with a spectral hydrostatic dynamical core. Additionally, other physical processes (turbulent diffusion, convection, clouds, precipitation, gravity wave drag, diabatic heating by radiation) are calculated for each vertical column of the Gaussian grid associated with the truncation used in the spectral dynamical core. These processes are coupled with the horizontal circulation each time step by transforming the variables that represent the atmospheric state from the spectral representation to the Gaussian grid and back. However, radiative transfer is computed extensively only once per hour for solar radiation (14 bands) as well as terrestrial radiation (16 bands).

As integral component of ECHAM6, JSBACH provides the lower atmospheric boundary conditions over land as well as biogeochemical and biogeophysical degrees of freedom that arise from terrestrial processes. JSBACH simulates land-surface properties interactively in terms of soil moisture, snow cover, leaf area index, and vegetation distribution. Plant diversity is represented in JSBACH in discrete functional plant classes, so-called "Plant Functional Types" (PFTs). The standard JSBACH set provides 21 PFTs representing natural vegetation, crops and pasture. The submodel for biogeographic vegetation shifts accounts for only 8 PFTs, shown in Tab. 1. Natural land cover change and vegetation dynamics are simulated in JSBACH by the DYNVEG component (Brovkin et al., 2009) based on a number of principles commonly used in Dynamic Global Vegetation Models (DGVMs), briefly summarized in the following. For a detailed description see Reick et al. (2013).

The "universal presence principle" implies that each PFT can potentially grow everywhere ("seeds are everywhere"). Physiological constraints define the climatic range within which a certain PFT can exist. Such bioclimatic limits only prohibit establishment if conditions are not suitable for a PFT to grow, but do not prevent further existence when values fall out of this range. The increase or reduction of land cover is determined by two processes. First, PFT cover can be reduced by natural death or disturbance (e.g. wildfires) and increased by migration into space opened in this way, so-called "uncolonized land". The different PFTs compete for this uncolonized land while vegetation establishment is generally only possible when net primary productivity ($NPP$) is positive at least for some years. Competition is considered in DYNVEG by growth form and by productivity. After disturbances, grass PFTs have an advantage because they quickly migrate in the new space while woody PFTs (trees and shrubs) regrow slowly. The tree-grass ratio depends on the rate of disturbances. Within the woody PFTs, competition is regulated by productivity: higher $NPP$ means a competitive advantage. PFTs with higher $NPP$ migrate faster into uncolonized land. In absence of disturbance, woody PFTs are dominant over grass PFTs by reason of light competition. The


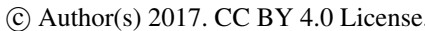


second possibility for land cover to increase or decrease is that inhospitable regions can expand or shrink. This change in area available for growth affects the cover of all PFTs.

The dynamic modelling of natural vegetation is based on fractions of unit area in a grid cell as the state variables in DYN-VEG. A composition of woody ($w_i$), grass ($g_i$), and uncolonized ($u$) cover fractions compose the whole area within a grid cell where vegetation cannot grow ($veg_{max}$):

$$u + \sum_{i=1}^{N^w} w_i + \sum_{i=1}^{N^g} g_i = 1 \ , \tag{1}$$

where $N^w$ and $N^g$ are the number of woody and grass PFTs, respectively. The dynamics of the cover fractions are governed by the coupled set of differential equations that account for establishment, natural mortality, and disturbances (fire, wind throw), acting on characteristic timescales for woody and for grass types (see Reick et al., 2013). The result is the potential natural vegetation cover in a world without humans. DYNVEG also includes a sophisticated approach to account for a human aspect, but we do not consider anthropogenic land cover change here.

The dynamics of the fraction of inhospitable land ($d = 1 - veg_{max}$) in a grid cell are calculated in DYNVEG with a separate submodel in order to determine the extent of cool deserts like the Arctic region or hot deserts like the Sahara. The extent of $d$ determines the fraction of a grid cell where vegetation can grow. The model is based on the idea that deserts develop when the long term $NPP$ average drops below a threshold so that vegetation cannot establish a canopy at least once a year. The fraction $f$ of a model grid cell with substantial vegetation cover at least once in year ($y$) is

$$f(y) = \sum_{i \in W} w_i (1 - e^{-a(LAI_i^{max})^b}) + \sum_{i \in G} g_i \frac{g + u}{g} (1 - e^{-a(LAI_i^{max})^b}) \ , \tag{2}$$

where $g = \sum_{i \in G} g_i$ is the total grass fraction of vegetation, and $LAI_i^{max}$ is the maximum leaf area that appeared during the year $y$. $LAI_i^{max}$ is determined from the maximum biomass in leaves by

$$LAI_i^{max}(y) = SLA_i \cdot C_{G,i}^{max}/3 \ , \tag{3}$$

where $C_{G,i}^{max}$ is the maximum living biomass found in PFT $i$ in the considered year. One third of the biomass is assumed to be in the leaves, and the specific leaf area ($SLA_i$) relates the carbon content of leaves to their area. The parameter $a = 1.95$ is chosen such that the simulated distribution of hot and cold deserts matches observations. The parameter $b = 2$ describes steepness of the transition between vegetation and desert and has been chosen to give a realistic distribution of deserts. Since one year of low growth does not make a desert, we assume a delayed response to changes in growth conditions where the time scale for desert development is chosen as 50 years.

## 2.2 Set up of simulations

We perform global coupled land-atmosphere simulations with a horizontal resolution of approximately 1.88° (T63) and 47 vertical levels. The model runs with dynamic vegetation for the periods 8 ky (ky = 1000 years before present), 6 ky, 4 ky, 2 ky, and 0 ky for 300 years with the first 200 years corresponding to a spin up period to reach a state close to equilibrium; the first





100 years of the spin up period run with three times accelerated vegetation dynamics. In the standard JSBACH configuration, the seasonal canopy albedo is calculated as a function of leaf area index, whereas the bare soil albedo is a grid box constant derived from satellite measurements (see Otto et al., 2011). To account for darker soils below vegetation, we implement for the study domain (12 to 34° N, -15 to 40° E, see Fig. 1) a simple albedo-scheme that reduces the soil albedo according to the mean

net primary productivity of the preceding five years (based on Vamborg et al. (2011)). Soil properties originate from the FAO digital soil map of the world (FAO/UNESCO, 1974). Due to lack of palaeo soil data, we are obliged to assume that the soil characteristics during the AHP were the same as they are today. To attain equilibrium states, we set orbital parameters (Berger, 1978) and $CO_2$ concentrations (Joos, 2016) to fixed values for each time slice experiment, see Tab. 2. Other atmospheric boundary conditions (trace gas concentrations, aerosol distribution, spectral solar irradiance, orography) remain unchanged

over time. Moreover, we prescribe sea ice concentration (SIC) and sea surface temperatures (SST) identically for all simulations using data from Hurrell et al. (2008) in cyclical repetition (1945 to 1974).

### 2.3 Modifications of plant functional diversity

To investigate the effects of changes in plant functional diversity on climate–vegetation interaction towards the end of the AHP, we perform four different types of simulations: as a baseline serves the simulation $EXP_{ALL}$ with all natural PFTs

commonly used in JSBACH (Tab. 1). The second type of simulation is a single-PFT experiment $EXP_{C4}$ which features only C4 Grass ($C4$) and thereby excludes all woody PFTs and their competition in the study domain. Third, we combine the woody PFTs Tropical Evergreen Tree ($TE$) and Raingreen Shrub ($SRG$) with C4 Grass in the study domain for the experiments $EXP_{TE,C4}$ and $EXP_{SRG,C4}$. The forth type of simulation addresses the representation of bioclimatic limits in JSBACH. During the experimental phase of the present study, we found that the minimum temperatures of the coldest month fall in some

regions in north Africa below the bioclimatic limit of Tropical Evergreen and Deciduous Trees (15.5°C) and thereby prevent their establishment in areas where reconstructions indicate the presence of tropical trees during the AHP (Hély et al., 2014). Although the definition of this limit is based on an empirical relationship between mean temperatures of the coldest month and absolute minimum temperature (frost occurrence) (Müller, 1982; Prentice et al., 1992), favorable microclimatic conditions could have allowed tropical taxa to establish. Another argument is that there might have been taxa that were partially frost

tolerant and could survive short periods of freezing temperatures. A literature-based compilation of experimental cold tolerance thresholds for leaves (evergreens), buds and twigs or stem illustrates that "tropical evergreens show damage at - 1°C or -2°C. Many of the broad-leaved evergreens can tolerate - 10 to -15°C, with a few able to survive -20°C" (Harrison et al., 2010). In order to test the model sensitivity to this limit, we add a newly designed tropical frost tolerant woody PFT ($TD_{10}$) with relaxed bioclimatic limits to the standard PFT set in $EXP_{TD10}$. The minimum temperature of the coldest month is for $TD_{10}$ reduced

from 15.5 to 10°C, all other parameters equate to the Tropical Deciduous Tree PFT ($TD$), see Tab. 1. In all simulations, the initial cover fractions are equally distributed over all included PFTs in the study domain.

In the following, we refer to the "potential" PFT diversity as the number of PFTs allowed in a simulation and to the "realized" PFT diversity as the number of PFTs that actually establish in a grid cell. We present the effects of changes in plant functional





diversity on 1) climate–vegetation interaction during the AHP and the extent of the "green" Sahara (8 ky), and on 2) the timing and rate of the transition from the "green" Sahara to the desert state.

## 3   Results

### 3.1   Effects on climate–vegetation interaction during the AHP

Changes in plant functional diversity significantly affect climate–vegetation interaction and the extent of the "green" Sahara under AHP boundary conditions (8 ky). Figure 2 illustrates precipitation ($P$) and vegetation cover fraction ($veg_{max}$) for the experiment with the standard PFT set ($EXP_{ALL}$) as well as differences between $EXP_{ALL}$ and simulations with modified PFT composition. Already at the first glance, it becomes apparent that $P$ and $veg_{max}$ considerably differ between experiments: $EXP_{C4}$, $EXP_{TE,C4}$, and $EXP_{TD10}$ depict in most of the study domain a "greening" associated with higher $P$ compared to

$EXP_{ALL}$ (Fig. 2c-f, i-j) while $EXP_{SRG,C4}$ shows a "browing" and less $P$ (Fig. 2g, h).

A closer look reveals substantial regional differences within each simulation. To identify the underlying causes, we compare land surface parameters, energy surface budget, and hydrological cycle in the two most affected regions (see Fig. 2): the transition zone between desert and savanna (Region 1: 18 to 22° N, 5 to 30° E) and the region southwest of the transition zone (Region 2: 12 to 18° N, -15 to 20° E). Tab. 3 summarizes selected parameters averaged over the last 100 simulated years

in the two regions for $EXP_{ALL}$ as well as differences to the other simulations. The fractions ($f_i$) of PFTs with a substantial contribution to the total vegetation cover fraction $veg_{max}$ are shown in Tab. 4. Further, we consider how changes in land surface parameters, surface energy balance, and hydrological cycle translate into alterations of large scale circulation features associated with the West African monsoon focusing on the lower-level Inner Tropical Convergence Zone (ITCZ, 925 hPa), the mid-level African Easterly Jet (AEJ, 600 hPa), and the upper-level Tropical Easterly Jet (TEJ, 150 hPa) during the monsoon

season (June to September) (Fig. 3). The term "ITCZ" is ambiguous since literature provides very different definitions based on wind convergence, surface air pressure and rainfall or outgoing longwave radiation. Hereinafter, we refer to the ITCZ as the surface feature over the African continent that marks the convergence of northeasterly Harmattan winds that originate in the Sahara and the southwest monsoon flow that emanates from the Atlantic, also named Inner Tropical Front.

In Region 1 in $EXP_{C4}$, $EXP_{TE,C4}$, and $EXP_{TD10}$, the lower albedo due to higher $veg_{max}$ compared to $EXP_{ALL}$

enhances the absorption of solar energy available for evapotranspiration which heats the lower atmosphere, accelerates energy and water cycling, and boosts moisture available in the atmosphere. This does not only increase the likelihood for regional convective precipitation, but also enhances the meridional temperature and moisture gradient between the equator and the northern Tropics. This gradient is a good indicator of how far north the monsoon flow penetrates into Africa (Bonfils et al., 2001). The increase of this gradient results in a northward shift of the ITCZ, a weakening and northward shift of the AEJ,

a strengthening of the TEJ, and consequently, a northward shift of the tropical rain belt (Fig. 3c-f, i-j). These changes are in accordance with literature: wetter than average conditions in the Sahel are linked to a weaker and northward shifted AEJ (Nicholson and Grist, 2003; Nicholson, 2013) and a stronger TEJ (Grist and Nicholson, 2001; Jenkins et al., 2005; Nicholson, 2008; Hulme and Tosdevin, 1989).

(c) Author(s) 2017. CC BY 4.0 License.



Complementary, the higher albedo in $EXP_{SRG,C4}$ compared to $EXP_{ALL}$ lowers the absorption of solar energy available for evapotranspiration which slows down water and surface energy fluxes, dries and cools the atmosphere, and suppresses precipitation in Region 1. The resulting reduction in the meridional atmospheric moisture and temperature gradient causes a southward shift of the ITCZ, the AEJ and the core of the rain belt (Fig. 3g, h). In agreement with literature, a more equator-

ward position of the AEJ is associated with drier than average conditions over the Sahel (Nicholson and Grist, 2003; Nicholson, 2013). However, the weakening of these two tropical jets is usually linked to wet conditions which is not the case in $EXP_{SRG,C4}$. This could be attributed to the low water recycling efficiency of vegetation compared to $EXP_{ALL}$, especially due to $SRG$, which implies less release of latent heat in the atmosphere which in turn decreases convection (Texier et al., 2000).

In Region 2 in $EXP_{C4}$ and $EXP_{TE,C4}$, and $EXP_{TD10}$, $veg_{max}$ varies only little with changes in PFT composition compared to $EXP_{ALL}$, but $P$ is largely affected by the properties of prevailing PFTs. In $EXP_{C4}$, the high grass albedo, compared to tropical tree PFTs that covers a substantial fraction of Region 2 in $EXP_{ALL}$ (Tab. 4), reduces the absorption of incoming solar radiation and surface energy fluxes. Additionally, the limited evapotranspiration capacity of the $C4$ PFT due to a comparatively small maximum leaf area ($LAI_{max,i}$) limits the transfer of interception and soil water to the atmosphere (Tab. 1).

Together with the small surface roughness of $C4$, which reduces turbulent fluxes, this decreases the likelihood of precipitation south of the tropical rain belt (Fig. 3c). Based on the assumption that less precipitation supports less plant growth, one would expect both $veg_{max}$ and $P$ to be lower in $EXP_{C4}$ than in $EXP_{ALL}$. However, the efficient growth of $C4$ compensates the lower $P$ relative to $EXP_{ALL}$. $C4$ requires less water to cover the same area with leaves than other PFTs (Fig. 4b) because $C4$ has a high $SLA_i$, a high $NPP$ (Tab. 1), and the photosynthetic C4 pathway enables a high water use efficiency. For example,

to cover 1 m$^2$ area with 1 m$^2$ leaves, $C4$ needs around 210% less $NPP$ than $TE$ and around 260 % less than $SRG$ (derived from Eq. (2),( 3)). Additionally, grass cover is not directly reduced by disturbances.

The moderate effects on precipitation in $EXP_{TE,C4}$ reflect the relatively small changes in land surface properties because the PFT composition is almost identical to $EXP_{ALL}$ (Tab. 4). In $EXP_{TD10}$, the increase in $P$ in Region 2 results mainly from a lower albedo, a higher surface roughness, and a higher evapotranspiration in Region 1 as well as in the northwestern

part of the study domain due to a higher contribution of tree PFTs which limits $SRG$ growth. This strongly enhances the West African monsoon (Fig. 3i,j) and thereby significantly increases rainfall over large areas of the study domain. Such an albedo induced enhancement of precipitation in the Sahel region has been shown in previous MPI-ESM experiments (Bathiany et al., 2010; Vamborg et al., 2011).

In $EXP_{SRG,C4}$, $veg_{max}$ is in Region 2 lower than in the other simulations with the same precipitation (see Fig. 2e, f, 4d).

The explanation lies in the parameterisation of $SRG$ and in the implementation of competition in JSBACH. Due to a low $SLA_i$ (Tab. 1), $SRG$ requires a higher $NPP$ to cover the same area with leaves as another PFT. $SRG$'s comparatively low photosynthetic capacity and low $LAI_{max,i}$ however impedes reaching a $NPP$ in the order of magnitude as other PFTs (Fig. 4d). When competition is calculated in JSBACH, $SRG$ outcompetes $C4$ due to the implicit assumption of light competition (see Sect. 2.1). Since growth conditions are not optimal for $SRG$ in the study domain facing water competition with $C4$ – all PFTs

use water from the same soil water reservoir – $SRG$ cannot fill the pools of living biomass over the growing season, which





leads to an expansion of desert area (see Eq. (2), (3)). $SRG$ thereby acts as a desert promoter in JSBACH. Complementary, the lack of $SRG$ in $EXP_{C4}$ and $EXP_{TE,C4}$ reduces the competitive pressure on $TE$ and $C4$ and their higher growth efficiency facilitates their expansion and the repression of desert. The desert promoting effect of shrubs has been observed in previous experimental studies ("fertile island" effect, e.g. Schlesinger et al. (1990, 1996); Whitford (2002)), but for other reasons – the

complex processes involved are not explicitly implemented in JSBACH.

Another important aspect to be considered is that various factors can affect $veg_{max}$ in coupled simulations apart from precipitation, which is the main determinant of plant growth in semi-arid regions on the considered scale of the order of a GCM grid cell (Coughenour and Ellis, 1993). Two important additional factors appear upon closer inspection of the outliers in Fig. 4. First, shallow soils in mountainous regions (in JSBACH, compare Fig. 1) are not capable of holding water and therewith

impede plant growth despite high precipitation. Second, in regions where temperatures of the coldest month fall below the bioclimatic threshold of $TE/TD$ (15.5° C), these tropical PFTs cannot establish thereby favoring the dominance of other PFTs with different water requirements. This effect is substantially reduced in $EXP_{TD10}$ with the implementation of a frost tolerant Tropical Tree PFT.

### 3.2    Effects on the transition from the "green" Sahara to the "desert" state

For a first estimate of regional transition patterns from the "green" Sahara (8 ky) to the "desert" state (0 ky), we subtract 100-year averages of consecutive time slices for all simulations and compare the resulting transition maps of $P$ and $veg_{max}$ in the whole study domain (12 to 34° N, -15 to 40° E). Note that this analysis does not represent the transient changes of vegetation extent and precipitation over the last 8000 years but provides an estimate of possible different states for a series of external forcings. This set up implies vegetation being permanently in equilibrium with climate. In reality, the delayed response

of vegetation allows several potential transient conditions to exist before diversity slowly attains equilibrium (Vellend et al., 2006; Diamond, 1972; Brooks et al., 1999). However, as the time scales of the simulated vegetation dynamics and atmospheric processes are much shorter than the 2 ky period between simulated time slices, this approach is legitimate for this study.

In $EXP_{ALL}$, the western part of the study domain experiences a stronger precipitation reduction than the eastern part in all periods (Fig. 5, left column). The latitudes of strongest precipitation decline shift gradually southward indicating a southward

shift of the tropical rain belt over time. A slight precipitation increase at the southwestern coast in the first period before precipitation starts decreasing in concert with the rest of the domain supports this indication. The magnitude of $P$ decline ranges from less than 50 mm yr$^{-1}$(2 ky)$^{-1}$ in the northern part of the domain to more than 200 mm yr$^{-1}$(2 ky)$^{-1}$ at the latitudes of maximum change between 12 and 20° N, with single grid cells in the western part reaching up to 250 mm yr$^{-1}$(2 ky)$^{-1}$. $veg_{max}$ follows the pattern of $P$ decline with a latitudinal offset of around one grid cell to the north, reaching maximum rates

of decrease from 0.1 to 0.2 yr$^{-1}$(2 ky)$^{-1}$ in the transition zone between desert and savanna (Fig. 5, right column). This transition zone shifts southward from 18 to 22° N at 8 ky to around 14 to 20° N at 0 ky. The latitudinal offset indicates that vegetation does not respond directly to changes in $P$ but declines when a threshold is reached at low precipitation rates. The almost constant rates of $P$ and $veg_{max}$ decline in the latitudes of maximum change in all periods represent a gradual transition from the "green" Sahara to the "desert" state.



The simulations with modified PFT diversity show qualitatively similar patterns of $P$ and $veg_{max}$ decline (see appendix). However, the timing and rate of transition substantially differ between simulations with different PFT compositions. For the quantitative comparison of all simulations, we condense the information of the transition maps by calculating zonal means (-15 to 40° E) of $P$ and $veg_{max}$, in the following referred to as $\Delta P$ and $\Delta veg_{max}$. Figure 6 illustrates that the maximum as
well as the temporal evolution of $\Delta P$ and $\Delta veg_{max}$ considerably differ between simulations.

Just as described above for the transition maps, the latitudes of maximum change shift in $EXP_{ALL}$ gradually southward by one grid cell per period (Fig. 6a). $\Delta veg_{max}$ follows the symmetric pattern of $\Delta P$ with a meridional offset of around one grid cell to the north, reaching maximum rates of decrease in the transition zone between desert and savanna (Fig. 6b). In $EXP_{C4}$, $EXP_{TE,C4}$, and $EXP_{TD10}$, the temporal evolutions of $\Delta P$ and $\Delta veg_{max}$ exhibit a delayed and sharpened transition
compared to $EXP_{ALL}$ (Fig. 6c-f, i-j), whereas $EXP_{SRG,C4}$ depicts a delayed but smoothed transition (Fig. 6g, h). $\Delta veg_{max}$ shows in $EXP_{SRG,C4}$ an exceptional behavior with a slow but non-monotonous southward shift of the most changing latitudes.

The explanation for differences in timing and magnitude of vegetation and precipitation decline between simulations with different PFT compositions is twofold. The first part of the explanation lies in dissimilar initial precipitation and vegetation cover fraction values. High initial values of $P$ and $veg_{max}$ in $EXP_{C4}$, $EXP_{TE,C4}$, and $EXP_{TD10}$ in the transition zone
between desert and savanna imply a large gradient between "green" and "desert" state and therewith inherit a larger potential for high $\Delta P$ and $\Delta veg_{max}$ than $EXP_{ALL}$ and $EXP_{SRG,C4}$ (Fig. 2, Tab. 3).

The second part of the explanation lies in the disparate relationships between precipitation and vegetation cover fraction and in the associated different sensitivities to precipitation decline that are specific to the particular PFT composition, depicted in the vegetation–precipitation diagrams (V–P diagrams) in Fig. 4 for all simulations at 8 ky (blue) and 0 ky (red) including all
grid cells in the study domain to cover the full precipitation spectrum.

If we consider a grid cell $j$ at the upper end of the dominant branch in any simulation, the qualitatively constant relationship between $P$ and $veg_{max}$ over time indicates that $veg_{max,j}$ "moves" left along the branch in the V–P diagram when $P_j$ declines. Within a certain precipitation range, $veg_{max,j}$ is not affected by a precipitation reduction until the threshold of maximum cover is reached. The position of this threshold is determined by the ratio of involved PFTs resulting from competition, their
productivities and capabilities to suppress desert expansion as described earlier in Sect.3.2. When $P_j$ drops below the threshold of maximum cover, $veg_{max,j}$ starts to decrease according to the slope of the branch. If the threshold value is low, the slope of the branch is steep, and $veg_{max,j}$ drops abruptly with a small precipitation decline. If the threshold value is higher, the slope is shallower, and $veg_{max,j}$ retreats more gradually. The regional response is determined by the dominant branch, therewith sharp in $EXP_{C4}$, $EXP_{TE,C4}$, and $EXP_{TD10}$, gradual in $EXP_{ALL}$, and very shallow in $EXP_{SRG,C4}$. With further precipitation
decline below the threshold, not only $veg_{max,j}$ decreases, but the PFT composition changes as well. PFTs with high moisture requirements cannot sustain growth and are consequently replaced by more drought resistant PFTs. The alteration in PFT composition implies that the relationship between $P_j$ and $veg_{max,j}$ changes in the respective grid cell, thus the grid cell value "jumps" to another branch and follows its trajectory with further precipitation decrease. This jump happens most obviously in $EXP_{SRG,C4}$. When $SRG$ disappears after 4 ky and thereby allows $C4$ to establish, affected grid cells shift to the upper branch



in the diagram, which resembles the main branch of $EXP_{C4}$, and reach a higher cover fractions with the same precipitation amount. This shift also explains the non-monotonous vegetation retreat in Fig. 6h.

## 4   Discussion

The present study is a first attempt to account for effects of plant functional diversity on climate–vegetation interaction in a comprehensive Earth system model. Our results confirm previous conceptual studies on the effect of plant diversity on climate–vegetation interaction (Claussen et al., 2013; Groner et al., 2015) in accordance with the "diversity-stability" hypothesis (McCann, 2000; Scherer-Lorenzen, 2005): high diversity can smooth the vegetation response to an externally forced precipitation decline, as we demonstrate in the comparison between $EXP_{ALL}$ and $EXP_{C4}/EXP_{TE,C4}$. Our findings thereby reconcile a gradual transition from a "green" state to a "desert" state with a strong feedback between single plant types and climate. On the other hand, the removal or introduction of key stone species – here $SRG$ or $TD_{10}$ respectively – can substantially alter the vegetation response to an externally forced precipitation decline. Despite the potential PFT diversity is highest in $EXP_{TD10}$, the properties of realized PFTs change the vegetation dynamics and the interaction with the atmosphere such that the transition happens with a similar fast rate as in the simulation with only one PFT ($EXP_{C4}$). In contrast, the dominance of the desert promoting $SRG$ in $EXP_{SRG,C4}$ leads to a transition that is even more gradual than in the experiments with higher potential PFT diversity. Thus, not the absolute number of potential PFTs, but the realized PFT composition determines climate–vegetation interaction and the system response to changing external forcing.

The sensitivity of the simulated system to changes in PFT composition and PFT properties could be an additional explanation why previous studies showed different extents of the "green" Sahara (see Claussen et al., 2017). Some land surface models such as ORCHIDEE (Krinner et al., 2005) do not have a Raingreen Shrub PFT which plays a crucial role in competition in JSBACH, especially in the transition zone between desert and savanna. Further, the definition of bioclimatic limits in terms of minimum temperature of the coldest month, which is set to 15.5°C in many land surface models, prevents tropical tree PFTs from establishment in regions where they were reconstructed for the AHP. With the significant increase of vegetation cover fraction north of 20°N, the simulation with a frost tolerant tropical tree $EXP_{TD10}$ reaches a closer match to reconstructions than previous studies. The final choice of this bioclimatic limit requires further investigation.

Eventually, our findings raise the question how plant functional diversity should generally be represented in land surface models to obtain a functionally realistic description of vegetation. The PFT concept is the most commonly used approach to represent vegetation in the current generation of DGVMs, but its validity has been extensively discussed over the last years. The representation of plant diversity with a static set of discrete PFT parameters does not cover the range of species categorized as one PFT and disregards phenotypic plasticity and trait variability (Van Bodegom et al., 2012; Wullschleger et al., 2014) which are often larger within PFTs than between PFTs (de Bello et al., 2011; Kattge et al., 2011). Alternative approaches to represent plant diversity consider the simulation of individual plants (e.g. LPJ-GUESS (Smith et al., 2001); aDGVM (Scheiter et al., 2013)), trail variability (e.g. JSBACH (Verheijen et al., 2013, 2015)), trait flexibility (e.g. JeDi-DGVM (Pavlick, 2012); aDGVM (Scheiter et al., 2013)), or operate based on the evolutionary optimality hypothesis (Wang et al., 2017). Nevertheless,





the PFT concept remains the current standard method to represent vegetation in land surface models, and we suggest that the uncertainties arising from the incomplete representation of plant diversity need to be taken into account in the interpretation of modelling studies.

## 5   Summary and Conclusions

In the present study, we have illustrated how variations in plant functional diversity affect climate–vegetation interaction towards the end of the AHP in coupled land-atmosphere simulations.

In experiments with AHP boundary conditions, the extent of the "green" Sahara varies considerably with changes in plant functional diversity. Differences in vegetation extent and PFT composition in turn alter land surface parameters, water cycling and the surface energy budget. These changes have not only local consequences but significantly affect large scale atmospheric

circulation patterns indicating a strong feedback between the terrestrial biosphere and the atmosphere. In contrast with the general hypothesis of a positive climate–vegetation feedback, we find that higher vegetation cover is not necessarily associated with higher precipitation but determined by the properties of the predominant PFTs. We demonstrate that the simulated climate–vegetation system state is highly sensitive to the implementation of these properties by the example of bioclimatic limits in terms of minimum temperature of the coldest month for the PFT "Tropical Deciduous Tree".

Towards the end of the AHP, modifications of PFT diversity significantly impact the timing and rate of transition to the "desert" state. While the simulations with the standard PFT set in MPI-ESM show a gradual decrease of precipitation and vegetation cover over time, variations in potential PFT diversity cause either a sharp decline of both variables or an even slower response to the external forcing depending on the realized PFT composition. The explanation lies in different initial precipitation and vegetation cover values as well as in different relationships between precipitation and vegetation cover fraction

that are specific to the particular PFT composition.

Recapitulatory, we identify the realized PFT diversity rather than the potential PFT diversity as the decisive factor for climate–vegetation feedback strength, vegetation extent, and the timing and rate of transition from the "green" Sahara to the "desert" state in MPI-ESM. Since climate–vegetation interaction is highly sensitive to the PFT composition and the model-specific PFT representation, we expect that the observed effects are not limited to the subtropics during the mid-Holocene, but could occur in different regions, especially in transition zones between different biomes, under different external forcings, in-

cluding recent and future climate change. This raises the question how realistically Earth system models can actually represent climate–vegetation interaction, considering the poor representation of plant diversity in the current generation of land surface models. However, as long as the processes shaping ecosystems are still not fully understood, it remains a challenge to set the criteria for an appropriate representation of plant functional diversity in land surface models.

*Acknowledgements.* We thank Victor Brovkin for internal review and fruitful discussions.

The article processing charges for this open-access publication were covered by the Max Planck Society.



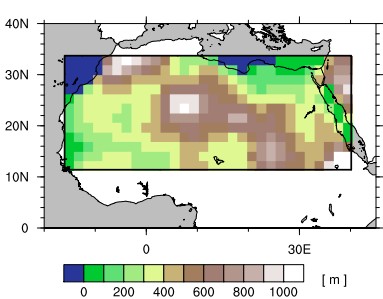

**Figure 1.** Mean orography (in m) of the simulated study domain in north Africa, 12 to 34° N, -15 to 40° E.





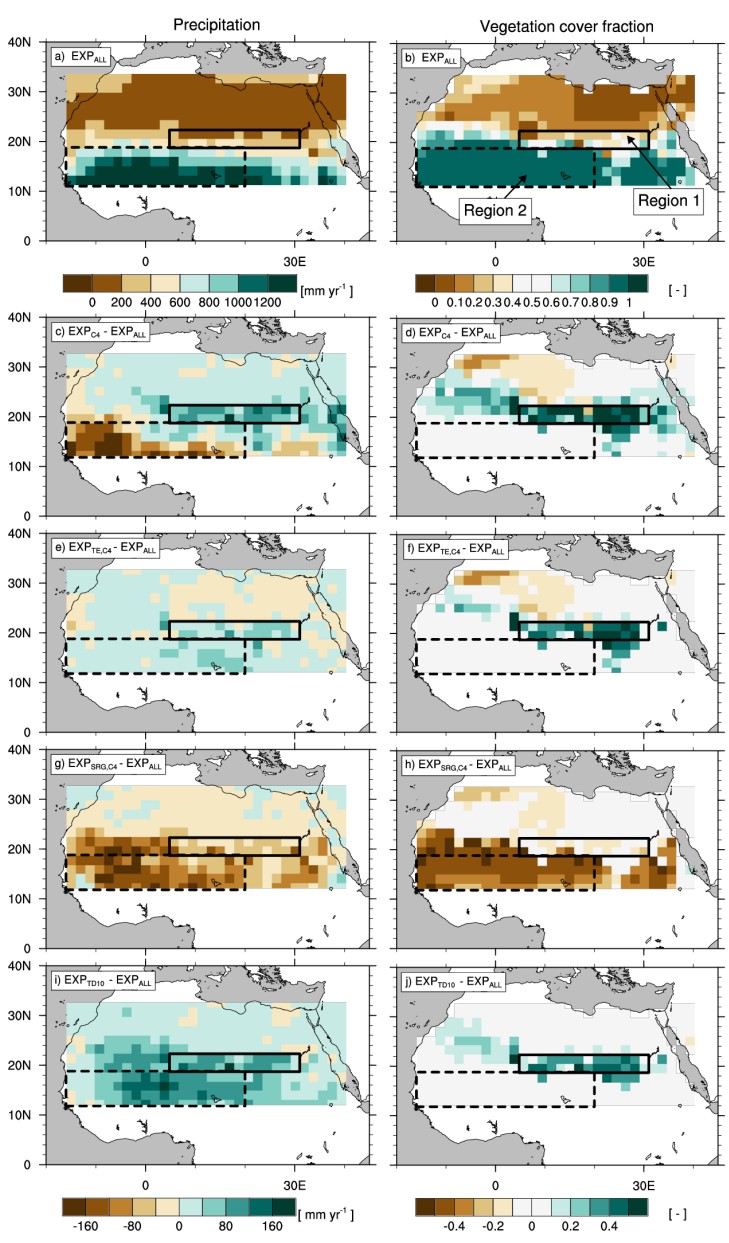

**Figure 2.** Effects of changes in plant functional diversity on precipitation $P$ (left column) and vegetation cover fraction $veg_{max}$ (right column) during the AHP (8 ky) in Region 1 (18 to 22° N, 5 to 30° E, solid box) and Region 2 (12 to 18° N, -15 to 20° E, dashed box). Panels (a, b) show 100-year averages for the experiment with the standard PFT set ($EXP_{ALL}$). The following panels illustrate differences in 100-year averages between $EXP_{ALL}$ and $EXP_{C4}$ (c, d), $EXP_{TE,C4}$ (e, f), $EXP_{SRG,C4}$ (g, h), and $EXP_{TD10}$ (i, j).





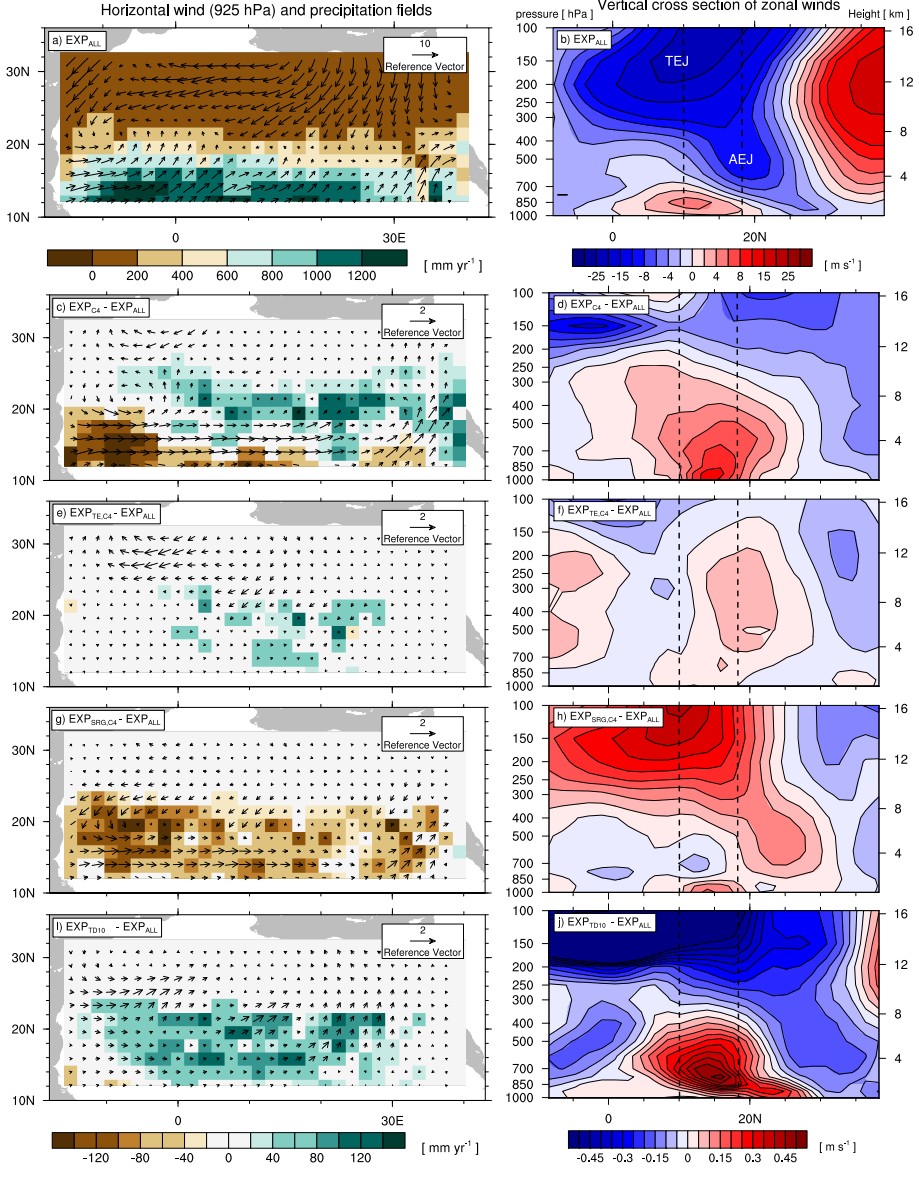

**Figure 3.** Effects of changes in plant functional diversity on precipitation $P$ and large scale atmospheric circulation patterns during the monsoon season (JJAS) under AHP conditions (8 ky). Left: horizontal low-level wind fields in the monsoon layer (925 hPa) and $P$ fields. Right: vertical cross sections of zonal winds (-10 to 40° N, -10 to 40° E). Dotted lines mark the core regions of mid-level African Easterly Jet (AEJ, 600 hPa) and upper-level Tropical Easterly Jet (TEJ, 150 hPa) in $EXP_{ALL}$. Panels (a, b) show the 100-year average for the experiment with the standard PFT set ($EXP_{ALL}$), the following panels illustrate differences in 100-year averages between $EXP_{ALL}$ and $EXP_{C4}$ (c, d), $EXP_{TE,C4}$ (e, f), $EXP_{SRG,C4}$ (g, h), and $EXP_{TD10}$ (i, j).





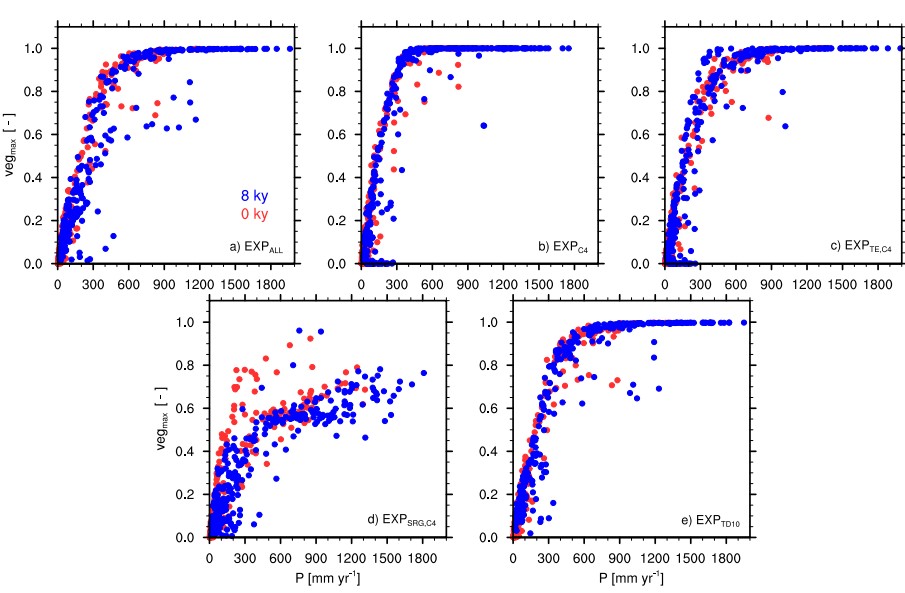

**Figure 4.** Vegetation–precipitation diagrams ($veg_{max}$, $P$) for simulations with different PFT combinations including all grid cells in the study domain (12 to 34° N, -15 to 40° E). Values are derived from 100-year averages under 8 ky (blue) and 0 ky (red) boundary conditions for $EXP_{ALL}$ (a), $EXP_{C4}$ (b), $EXP_{TE,C4}$ (c), $EXP_{SRG,C4}$ (d), and $EXP_{TD10}$ (f) .





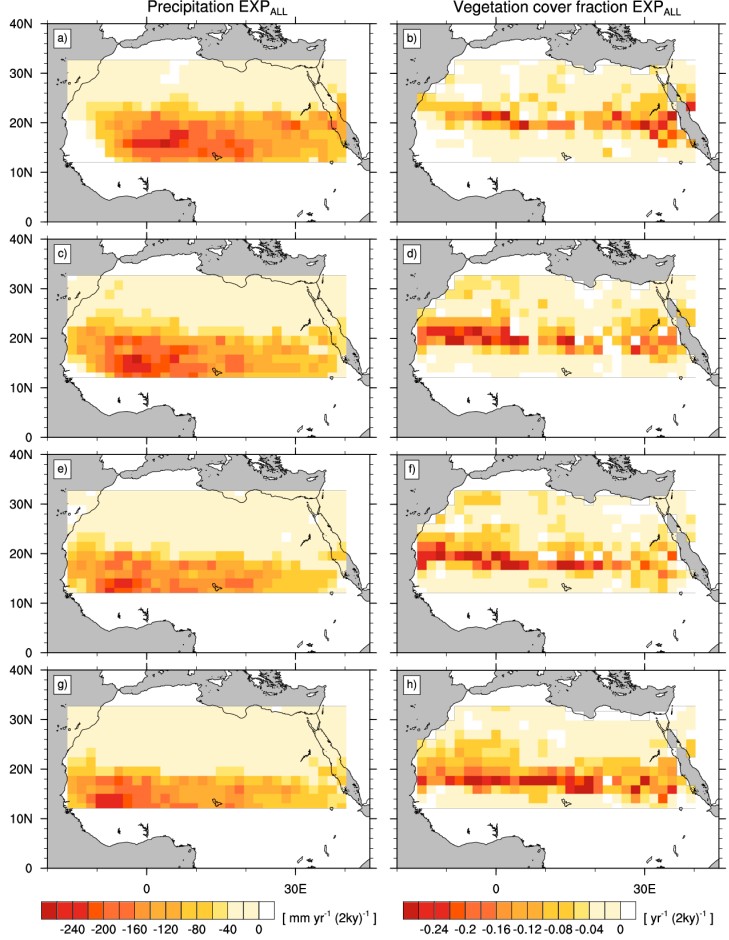

**Figure 5.** Transition rates from the "green" Sahara to the "desert" state for precipitation $P$ (left column) and vegetation cover fraction $veg_{max}$ (right column) of a simulation with the standard PFT set ($EXP_{ALL}$). Plots depict differences between consecutive time slices (100-year averages): 6 ky-8 ky (a, b), 4 ky-6 ky (c, d), 2 ky-4 ky (e, f), and 0 ky-2 ky (g, h).





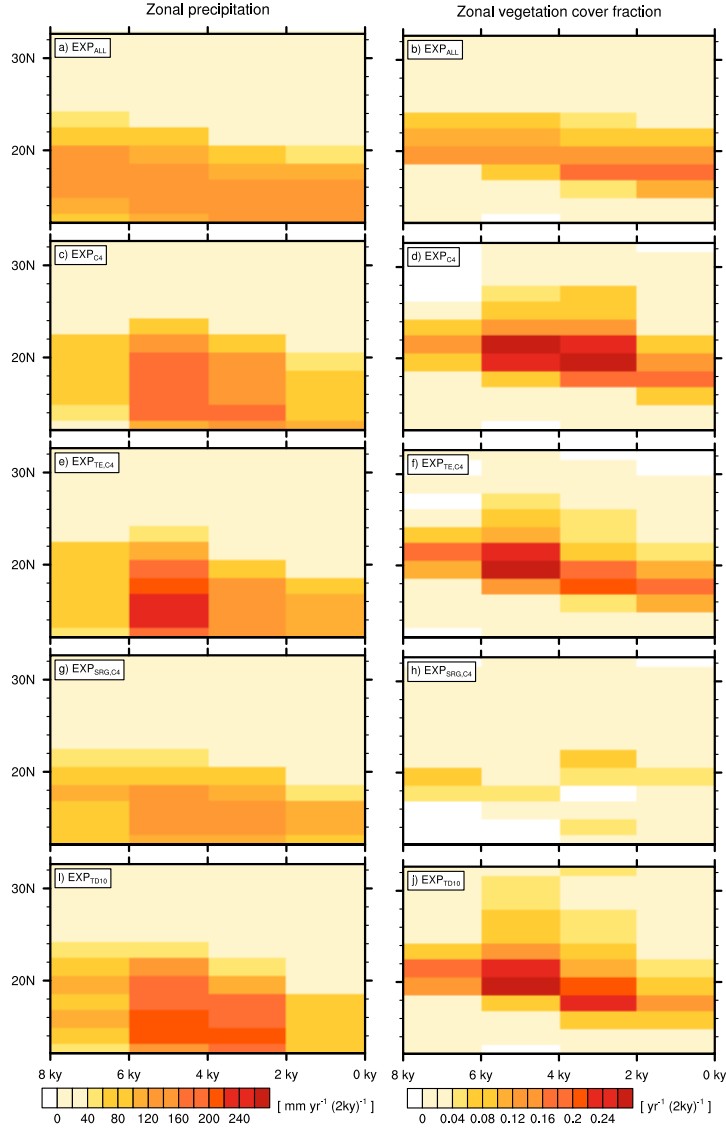

**Figure 6.** Zonally averaged transition rates from the "green" Sahara to the "desert" state for precipitation $\Delta P_i$ (left column) and vegetation cover fraction $\Delta veg_{max,i}$ (right column). Plots depict differences between consecutive time slices (100-year averages) for $EXP_{ALL}$ (a, b), $EXP_{C4}$ (c, d), $EXP_{TE,C4}$ (e, f), $EXP_{SRG,C4}$ (g, h), and $EXP_{TD10}$ (i, j).





**Table 1.** Natural Plant Functional Types in JSBACH, their woodiness type, associated time constants for establishment/mortality ($\tau_i$, in years), maximum carboxylation capacities ($V_{max,0,i}$) and electron transport capacities ($J_{max,0,i}$) at 25 °C (in $\mu$mol(CO$_2$) m$^{-2}$ s$^{-1}$), specific leaf area ($SLA_i$, in m$^2$(leaf) mol$^{-1}$(Carbon)), and maximum leaf area index ($LAI_{max,i}$, in m$^2$ m$^{-2}$) (Reick et al., 2013).

| Plant Functional Type | ID | type | $\tau_i$ | $V_{max,0,i}$ | $J_{max,0,i}$ | $SLA_i$ | $LAI_{max,i}$ |
|---|---|---|---|---|---|---|---|
| Tropical Evergreen Tree | $TE$ | woody | 30 | 39 | 74.1 | 0.264 | 7 |
| Tropical Deciduous Tree | $TD$ | woody | 30 | 31 | 59.8 | 0.376 | 7 |
| Extra-trop. Evergreen Tree | $eTE$ | woody | 60 | 44 | 83.6 | 0.110 | 5 |
| Extra-trop. Deciduous Tree | $eTD$ | woody | 60 | 66 | 125.4 | 0.304 | 5 |
| Raingreen Shrub | $SRG$ | woody | 12 | 61.7 | 117.2 | 0.184 | 2 |
| Deciduous Shrub | $SD$ | woody | 24 | 54 | 102.6 | 0.307 | 2 |
| C3 Grass | $C3$ | grass | 1 | 78.2 | 148.6 | 0.451 | 3 |
| C4 Grass | $C4$ | grass | 1 | 8 | 140 | 0.451 | 3 |

**Table 2.** CO$_2$ concentrations (in ppm) and orbital parameters for simulated time slices at 8 ky, 6 ky, 4 ky, 2 ky, and 0 ky. For palaeo simulations, CO$_2$ concentrations are taken from Joos (2016) and orbital parameters are adjusted according to Berger (1978). The values for 0 ky conform to the standard preindustrial set up of MPI-ESM.

| time slice | CO$_2$ [ppm] | eccentricity [ - ] | obliquity [ - ] | longitude of perihelion [ ° ] |
|---|---|---|---|---|
| 8 ky | 259.9 | 0.019101 | 24.209 | 148.58 |
| 6 ky | 264.6 | 0.01867 | 24.101 | 181.75 |
| 4 ky | 273.2 | 0.018123 | 23.922 | 215.18 |
| 2 ky | 277.6 | 0.017466 | 23.694 | 248.93 |
| 0 ky | 284.725 | 0.016704 | 23.44 | 283.01 |





**Table 3.** Effects of different PFT compositions on land surface parameters, surface energy budget, and hydrological cycle in coupled land-atmosphere simulations in Region 1 (18 to 22° N, 5 to 30° E) and Region 2 (12 to 18° N, -15 to 20° E). The first two columns contains 100-year averages of the experiment with the standard PFT set $EXP_{ALL}$, the following show differences between $EXP_{ALL}$ and simulations with modified PFT composition (Tropical Evergreen Tree $TE$, Raingreen Shrub $SRG$, C4 Grass $C4$, frost tolerant Tropical Deciduous Tree $TD_{10}$). Bold values are significant ($\sigma = 0.05$) with regard to the time series of 100 years (yearly averages), values in brackets correspond to spatial standard deviations of difference fields.

| | $EXP_{ALL}$ | | $EXP_{C4} - EXP_{ALL}$ | | $EXP_{TE,C4} - EXP_{ALL}$ | | $EXP_{SRG,C4} - EXP_{ALL}$ | | $EXP_{TD10} - EXP_{ALL}$ | |
|---|---|---|---|---|---|---|---|---|---|---|
| | Region 1 | Region 2 | Region 1 | Region 2 | Region 1 | Region 2 | Region 1 | Region 2 | Region 1 | Region 2 |
| *Land surface parameters* | | | | | | | | | | |
| Vegetation cover fraction [ - ] | 0.40(0.22) | 0.98(0.04) | **0.44**(0.21) | **0.02**(0.04) | **0.34**(0.21) | **0.01**(0.04) | **-0.12**(0.13) | **-0.40**(0.08) | **0.31**(0.16) | **0.01**(0.04) |
| Leaf area index [m² (leaf) m⁻² (canopy)] | 1.33(0.43) | 3.00(0.7) | **0.18**(0.33) | **-0.43**(0.38) | 0.03(0.32) | 0.02(0.04) | **-0.30**(0.17) | **-1.11**(0.57) | **0.15**(0.15) | **0.09**(0.06) |
| Albedo [ - ] | 0.28(0.07) | 0.17(0.01) | **-0.07**(0.06) | **0.02**(0.01) | **-0.06**(0.05) | **-0.001**(0.001) | **0.03**(0.04) | **0.01**(0.02) | **-0.07**(0.06) | **-0.001**(0.001) |
| Roughness length [m] | 0.08(0.07) | 1.32(0.69) | **-0.03**(0.07) | **-1.24**(0.68) | 0.03(0.09) | **0.07**(0.09) | **-0.04**(0.06) | **-1.19**(0.64) | **0.13**(0.07) | **0.11**(0.10) |
| *Surface energy budget* | | | | | | | | | | |
| Solar net radiation [W m⁻²] | 275.26(14.03) | 289.79(3.95) | **12.33**(12.00) | **-2.62**(1.46) | **10.78**(10.37) | 0.005(0.34) | **-4.76**(7.24) | 1.11(4.24) | **11.55**(11.10) | **-0.85**(0.55) |
| Sensible heat flux [W m⁻²] | 61.88(10.55) | 47.18(17.15) | **6.38**(11.00) | **-1.36**(3.82) | **7.17**(9.10) | -0.32(0.68) | **-3.77**(6.26) | 2.54(5.23) | **9.90**(10.42) | **-2.09**(1.1) |
| Latent heat flux [W m⁻²] | 20.07(7.88) | 70.12(22.70) | **5.21**(2.88) | **-6.27**(4.75) | **2.17**(2.29) | 0.53(0.88) | **-4.31**(2.74) | **-7.63**(3.72) | **5.64**(2.26) | **2.41**(1.31) |
| 2 m temperature [°C] | 24.95(1.54) | 26.21(1.42) | **0.46**(0.28) | **0.47**(0.21) | **0.50**(0.29) | 0.04(0.07) | -0.07(0.11) | **0.73**(0.28) | **0.37**(0.29) | -0.07(0.08) |
| Cloud cover [ - ] | 0.35(0.04) | 0.53(0.06) | 0.01(0.002) | **-0.02**(0.01) | 0.003(0.002) | -0.002(0.004) | **-0.02**(0.003) | **-0.03**(0.003) | 0.02(0.005) | 0.01(0.004) |
| *Hydrological cycle* | | | | | | | | | | |
| Precipitation [mm yr⁻¹] | 275.58(104.64) | 1140.86(352.69) | **84.87**(34.72) | **-76.19**(83.79) | **35.24**(28.97) | 24.81(17.03) | **-56.11**(33.16) | **-110.25**(45.96) | **82.73**(33.88) | **82.88**(40.40) |
| Evapotranspiration [mm yr⁻¹] | 253.31(99.45) | 884.82(286.44) | **65.76**(36.30) | **-79.16**(59.91) | **27.50**(28.84) | 6.69(11.10) | **-54.33**(34.55) | **-96.23**(46.88) | **71.29**(28.56) | **30.37**(16.58) |
| Integrated water vapor [kg m⁻²] | 20.49(1.86) | 32.62(3.52) | **0.62**(0.09) | **-0.40**(0.47) | 0.09(0.08) | 0.09(0.09) | **-0.95**(0.24) | **-1.40**(0.16) | **0.84**(0.30) | **0.73**(0.23) |





**Table 4.** Mean cover fractions $f_i$ (per vegetated area of a grid cell) of all Plant Functional Types (PFTs) with substantial shares of the total vegetation cover in Region 1 (18 to 22° N, 5 to 30° E) and Region 2 (12 to 18° N, -15 to 20° E). Values represent 100-year averages of the experiment with the standard PFT set $EXP_{ALL}$ and simulations with modified PFT composition (Tropical Evergreen Tree $TE$, Raingreen Shrub $SRG$, C4 Grass $C4$, frost tolerant Tropical Deciduous Tree $TD_{10}$).

| PFT | $EXP_{ALL}$ Region 1 | $EXP_{ALL}$ Region 2 | $EXP_{C4}$ Region 1 | $EXP_{C4}$ Region 2 | $EXP_{TE,C4}$ Region 1 | $EXP_{TE,C4}$ Region 2 | $EXP_{SRG,C4}$ Region 1 | $EXP_{SRG,C4}$ Region 2 | $EXP_{TD10}$ Region 1 | $EXP_{TD10}$ Region 2 |
|---|---|---|---|---|---|---|---|---|---|---|
| $TE$ | 0.01 | 0.40 | - | - | 0.05 | 0.42 | - | - | 0.01 | 0.42 |
| $TD$ | 0.02 | 0.01 | - | - | - | - | - | - | <0.01 | <0.01 |
| $TD10$ | - | - | - | - | - | - | - | - | 0.20 | <0.01 |
| $SRG$ | 0.14 | 0.01 | - | - | - | - | 0.18 | 0.34 | 0.01 | <0.01 |
| $C4$ | 0.20 | 0.57 | 0.84 | 1.0 | 0.69 | 0.57 | 0.1 | 0.25 | 0.48 | 0.55 |



## Appendix A

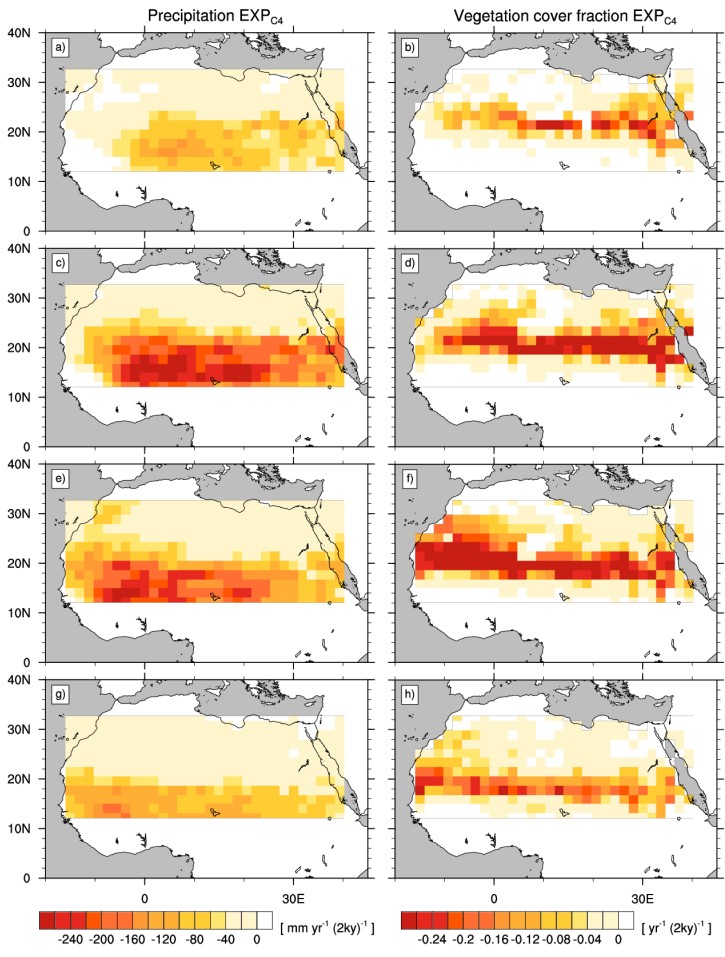

**Figure A1.** Transition rates from the "green" Sahara to the "desert" state for precipitation $P$ (left column) and vegetation cover fraction $veg_{max}$ (right column) of a simulation with C4 Grass only ($EXP_{C4}$). Plots depict differences between consecutive time slices (100-year averages): 6 ky-8 ky (a, b), 4 ky-6 ky (c, d), 2 ky-4 ky (e, f), and 0 ky-2 ky (g, h).



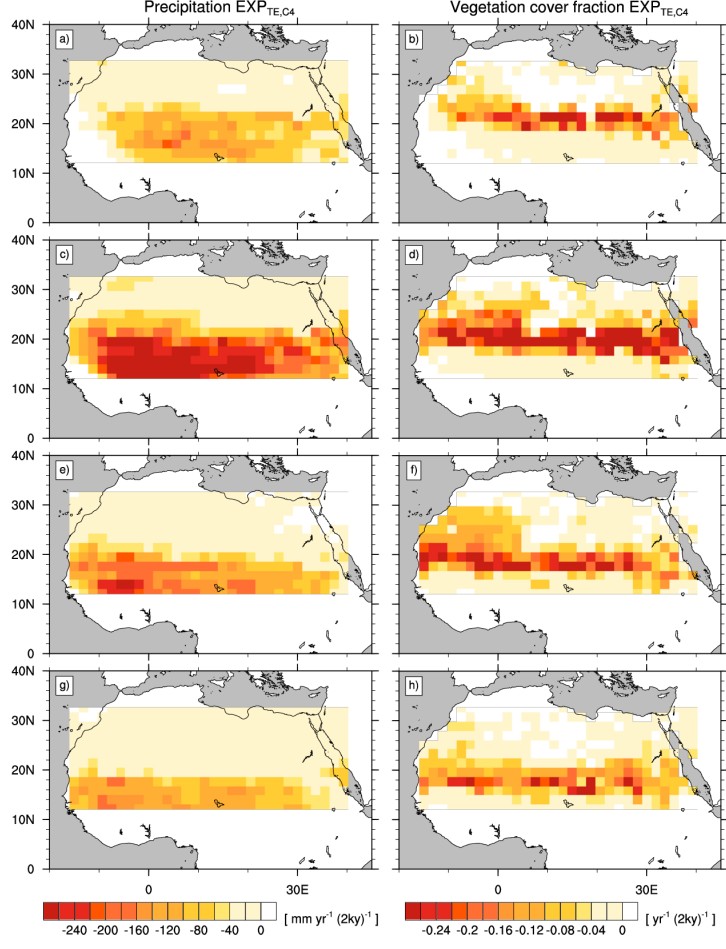

**Figure A2.** Transition rates from the "green" Sahara to the "desert" state for precipitation $P$ (left column) and vegetation cover fraction $veg_{max}$ (right column) of a simulation with C4 Grass and Tropical Evergreen Tree ($EXP_{TE,C4}$). Plots depict differences between consecutive time slices (100-year averages): 6 ky-8 ky (a, b), 4 ky-6 ky (c, d), 2 ky-4 ky (e, f), and 0 ky-2 ky (g, h).





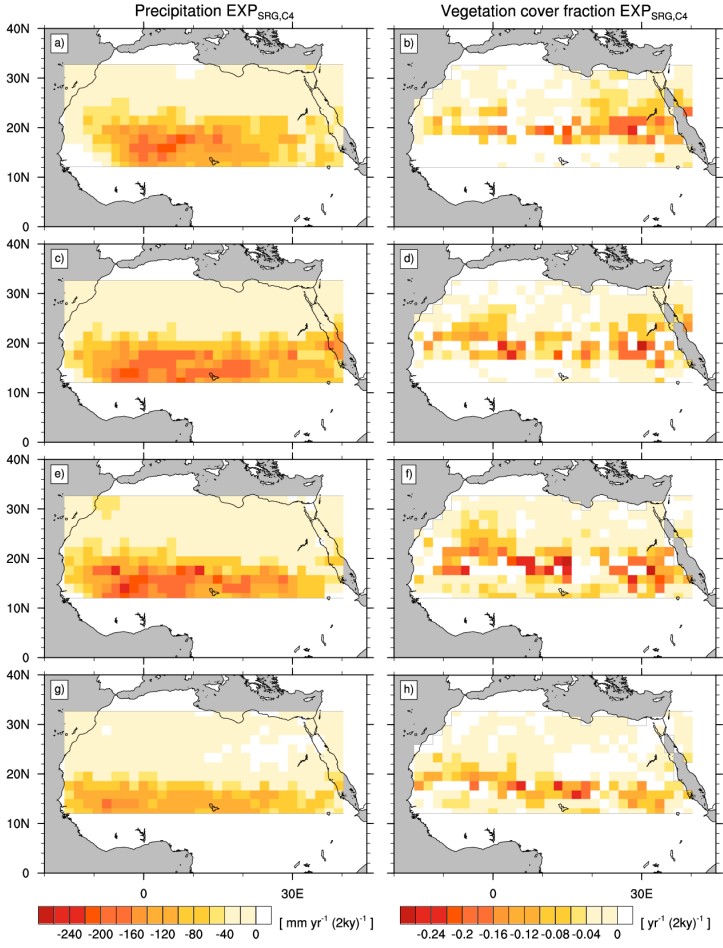

**Figure A3.** Transition rates from the "green" Sahara to the "desert" state for precipitation $P$ (left column) and vegetation cover fraction $veg_{max}$ (right column) of a simulation with C4 Grass and Raingreen Shrub ($EXP_{SRG,C4}$). Plots depict differences between consecutive time slices (100-year averages): 6 ky-8 ky (a, b), 4 ky-6 ky (c, d), 2 ky-4 ky (e, f), and 0 ky-2 ky (g, h).



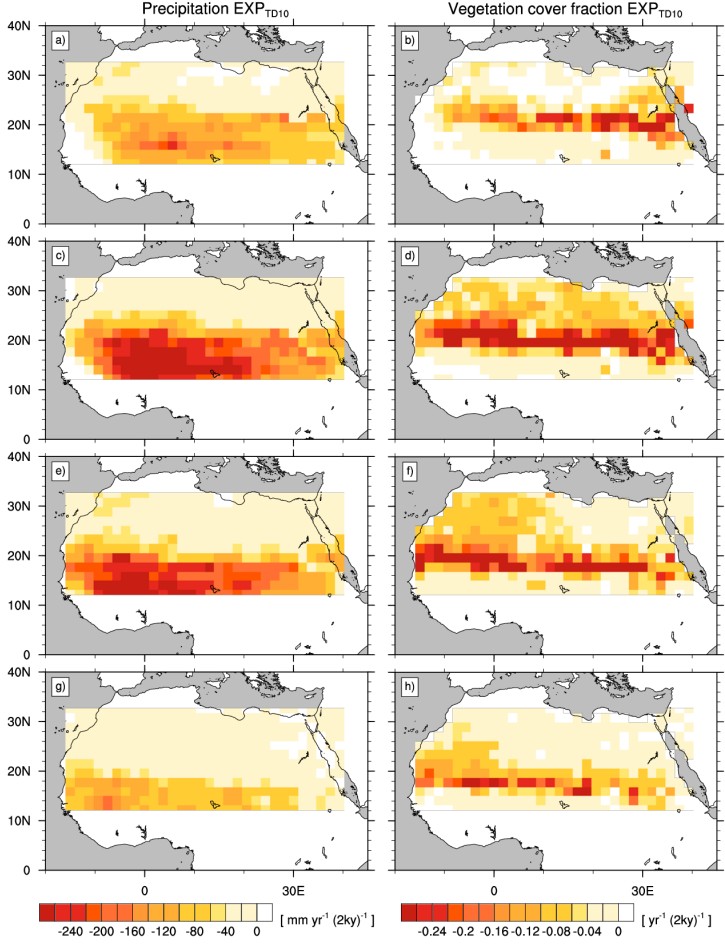

**Figure A4.** Transition rates from the "green" Sahara to the "desert" state for precipitation $P$ (left column) and vegetation cover fraction $veg_{max}$ (right column) of a simulation with the standard PFT set and a newly designed frost tolerant Tropical Deciduous Tree PFT ($EXP_{TD10}$). Plots depict differences between consecutive time slices (100-year averages): 6 ky-8 ky (a, b), 4 ky-6 ky (c, d), 2 ky-4 ky (e, f), and 0 ky-2 ky (g, h).



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
