# Peer review of "Plant functional diversity affects climate-vegetation interaction"

_Biogeosciences, 2017_

## Referee Comment (RC1) · Anonymous Referee #1 · 22 Dec 2017

The manuscript submitted by Groner et al to " Climate of the Past" discusses the influence of variations in plant functional types on climate at the end of the Holocene. The article is nicely written. However, it is based on assumptions that should be more fully argued or explained.

It has been shown for many years (actually, since the palaeogeographic map drawn up by N. Petit Maire et al., UNESCO-IGCP 252, 1993 and more recently through the reconstructions performed by e.g. Hély and Lézine, 2014) that plant cover in the Sahara during the Holocene was primarily not homogeneous with co-occurrence of plants that are today found in distinct phytogeographical areas. Desert plants are typically found in the Sahara today. These plants co-occurred during the late Holocene with tropical trees probably in restricted areas such as river or lake banks. Actually, pollen data

show that tropical trees were present but we are unable to infer any evaluation of their coverage in the landscape from pollen data. In other words, palynologists show that the Holocene increased rainfall led to a dramatic increase in biodiversity. There has been no replacement of one biome by another but rather an interpenetration of plants taxa that are found today in distinct biomes. Moreover, the vegetation cover had a mosaic-like character and was certainly discontinuous. An quantified evaluation of the vegetation cover would certainly be possible by applying the algorithms developed by Sugita and colleagues (2007) in West Africa. The importance of the coverage of one biome with respect to another one seems to me an important parameter to take into account, particularly if tropical trees were mostly restricted to the edges of streams and lakes. One of the major characteristics of plant distributions in dry areas is the presence of gallery forests along rivers or open water surfaces.

These gallery forests can host tropical trees far from the climatic zone they originate from. In this case, trees are not in equilibrium with climate and survive under drier conditions only thanks to available ground waters. Actually we do not know since when the water available in the soil is no longer able to compensate for the lack of precipitation.

In this context, could you please precise what are the 21 PFTs used in your study, based on the plant types identified in pollen studies carried out in the Sahara and Sahel and how do you evaluate the coverage and distribution of each of them for the time periods you have selected.

Additional comments:

(1) Temperature of the coldest month: To my knowledge, tropical climates are characterized by relatively constant (hot) temperature throughout the year and a large diurnal amplitude. One of the most important factors for plant distributions is rainfall and the length of the dry season, not temperature (at least in the lowlands). This, of course, in the case of a "climatic" and not an azonal distribution as is the case of forest galleries

(2) C3/C4 grasses: roughly 30% of the Poaceae growing in the Sahara today are C3, particularly those growing in wet places and in the highlands (Maire & Monod, 1950; Quézel, 1965; Maire, 1952; Quézel & Santa, 1962; Quézel, 1954; Gillet, 1968....)

---

## Author Comment (AC1) · 15 Jan 2018

Dear Referee #1, thank you very much for your comment. In order to consider your comment, we have to check back here if you actually refer to Groner et al. (2015) published in "Climate of the Past", or if the first part of your comment concerns the present manuscript, submitted to "Biogeosciences".

---

## Author Comment (AC2) · 18 Jan 2018

We thank the Anonymous Referee #1 for the insightful comments. In the following we will provide a detailed response.

**"The manuscript submitted by Groner et al to "Climate of the Past" discusses the influence of variations in plant functional types on climate at the end of the Holocene. The article is nicely written. However, it is based on assumptions that should be more fully argued or explained."**

In order to consider this comment, we have to check back here if the Referee actually refers to Groner et al. (2015) published in Climate of the Past, or if the first part of the comment concerns the present manuscript, submitted to Biogeosciences. For now we

assume that it is referred to the present manuscript.

**"It has been shown for many years (actually, since the palaeogeographic map drawn up by N. Petit Maire et al., UNESCO-IGCP 252, 1993 and more recently through the reconstructions performed by e.g. Hély and Lézine, 2014) that plant cover in the Sahara during the Holocene was primarily not homogeneous with co-occurrence of plants that are today found in distinct phytogeographical areas. Desert plants are typically found in the Sahara today. These plants co-occurred during the late Holocene with tropical trees probably in restricted areas such as river or lake banks. Actually, pollen data show that tropical trees were present but we are unable to infer any evaluation of their coverage in the landscape from pollen data. In other words, palynologists show that the Holocene increased rainfall led to a dramatic increase in biodiversity. There has been no replacement of one biome by another but rather an interpenetration of plants taxa that are found today in distinct biomes. Moreover, the vegetation cover had a mosaic-like character and was certainly discontinuous."**

We fully agree with this summary of the vegetation distribution during the African Humid Period. We based our study mainly on Hély et al. (2014) and their description of a highly diverse mosaic-like vegetation cover during the African Humid Period, see page 1 line 18. We agree that special features such as gallery forests must be described in more detail and we gladly introduce an additional descriptive paragraph to the introduction.

**"An quantified evaluation of the vegetation cover would certainly be possible by applying the algorithms developed by Sugita and colleagues (2007) in West Africa."**

As mentioned at the end of the introduction (page 2, line 29-31), "With our idealized set up, we do not expect our simulations to match reconstructions, rather we focus on qualitative differences between simulations to find mechanisms relevant for the question how PFT diversity affects climate–vegetation interaction."

**"The importance of the coverage of one biome with respect to another one seems to me an important parameter to take into account, particularly if tropical trees were mostly restricted to the edges of streams and lakes. One of the major characteristics of plant distributions in dry areas is the presence of gallery forests along rivers or open water surfaces. These gallery forests can host tropical trees far from the climatic zone they originate from. In this case, trees are not in equilibrium with climate and survive under drier conditions only thanks to available ground waters. Actually we do not know since when the water available in the soil is no longer able to compensate for the lack of precipitation."**

As mentioned above, we agree that gallery forests should be mentioned in the introduction as they are a crucial part of the study. $EXP_{TD10}$ with the additional frost-tolerant tropical tree PFT aims at representing gallery forests as far as possible in JSBACH, not spatially but conceptually. We understand that further description is necessary here and we gladly adjust the manuscript to make the intention behind this experiment more clear.

**"In this context, could you please precise what are the 21 PFTs used in your study, based on the plant types identifyed in pollen studies carried out in the Sahara and Sahel and how do you evaluate the coverage and distribution of each of them for the time periods you have selected."**

JSBACH provides in general 21 PFTs based on the most common land cover types, but we used only 8 PFTs in this study which are summarized in Tab. 1. The cover fraction of each PFT is calculated by the model as a result of climatic conditions and competition as described in the manuscript Section 2.1.

**"Additional comments: (1) Temperature of the coldest month: To my knowledge, tropical climates are characterized by relatively constant (hot) temperature throughout the year and a large diurnal amplitude. One of the most important**

**factors for plant distributions is rainfall and the length of the dry season, not temperature (at least in the lowlands). This, of course, in the case of a "climatic" and not an azonal distribution as is the case of forest galleries"**

We agree on that, however during the mid-Holocene winter temperatures were cooler in North Africa than today, and our model simulated temperatures below the bioclimatic threshold, thus tropical trees could not establish in the experiments. With the experiment $EXP_{TD10}$ we introduced the aspect of favorable microclimatic conditions especially in gallery forests.

**"(2) C3/C4 grasses: roughly 30% of the Poaceae growing in the Sahara today are C3, particularly those growing in wet places and in the highlands (Maire Monod, 1950; Quézel, 1965; Maire, 1952; Quézel Santa, 1962; Quézel, 1954; Gillet, 1968....)"**

The vegetation distribution presented in this study is a result of model simulations. We observe the occurrence of C3 grasses especially in the Mediterranean and in highlands, however under the given boundary conditions, C4 grasses are in our model more productive than C3 and thus the dominant grass PFT in most of the study domain.

---

## Referee Comment (RC2) · Anonymous Referee #2 · 26 Jan 2018

This manuscript aims to assess the sensitivity of the simulated extent of green Sahara during the African Humid Period (AHP), and the manner in which the land cover from this green Sahara transitions to current desert conditions , to the representation of plant functional types (PFTs) in the MPI-ESM (i.e. the PFT diversity). The subject of the manuscript is of broad interest and many past studies have addressed the question of the extent of green Sahara. Authors claim that differences in past studies are due to representation of different PFTs in different models (this appears to be a valid argument based on results presented in this manuscript) but the general strength of the land-atmosphere feedbacks (which depends on a particular land surface model) likely also plays a role. That is, the extent of green Sahara will be different amongst models despite same PFTs because different land surface models have different strengths of

land atmosphere coupling. In addition, different land surface models will likely represent same PFTs with different parameterizations and parameter values. This aspect is not discussed in the manuscript.

This study can be assessed in the context of paleo vegetation as the first reviewer appears to have done but also in the context of land-atmosphere interactions which is where my expertise lies. In my opinion, the manuscript needs clarification of several items before it may be considered for publication in BG.

The competition between PFTs is an important aspect of this study but I felt that the description on page 4 (lines 5 – 25) wasn't sufficient for me as a reader to understand how competition actually works. Since this is a modelling study, I feel it is important to lay it out for the reader. The text on page 5 attempts to do this but it seems it doesn't go all the way.

Other comments

Having read the authors' response to Reviewer #1 I now realize out of 21 PFTs only 8 PFTs can grow in the region considered. However, when I read the manuscript the first time I had similar confusion. So perhaps this point needs to be clarified.

Section 2.2 needs more info about model setup and discussion of implications of how this set up is done.

1) How does use of present day SSTs affects the overall results? Does an interactive ocean generally amplify or dampen the effect of land-atmosphere feedbacks?

2) Other atmospheric boundary conditions correspond to what time period?

3) What do soil properties mean – do you mean soil texture and permeable soil depth.

4) Since land-atmosphere feedbacks are key to understanding the results presented in this manuscript it would be useful to put albedo, typical LAI, rooting depth and vegetation height of different PFTs in a table for reader to understand how the different physical characteristics of a PFT can potentially affect land-atmosphere interactions.

On Page 6, around line 24, it is mentioned albedo in the C4 grass simulation is lower than that in the EXP_ALL simulation. Is this an error? Grasses are generally brighter than trees. But then on page 7, lines 11 and 12, the manuscript correctly notes that albedo is higher in EXP_C4 than in EXP_ALL. Please correct the sentences on page 6 so that they are consistent with the text on page 7.

On Page 7, line 2, the phrase "slows down water and energy fluxes" is unclear. What does slowing down means does it mean decrease in fluxes. If yes, which fluxes – I suppose evapotranspiration (i.e. latent heat). Please be explicit.

Page 7, line 16-20. Yes, C4 grasses are more productive than C3 grasses. But productivity is different than specific leaf area (SLA, m2/kg C) which is a measure of how many m2 of leaves can be constructed with a Kg of carbon of leaf biomass. The discussion in lines 16-20 appears to be mixing productivity with SLA.

Page 7, line 21. "Grass cover is not directly reduces by disturbances". This seems contradictory to what happens in nature. Grasses are more flammable than trees so fires affect grass cover more drastically – although, of course, grasses spring back faster too.

Page 7, line 33. "... SRG outcompetes C4 due to the implicit assumption of light competition". So are shrubs assumed to be taller than trees. This is where a more complete description of how competition works can help. A model can simulate the actual physical processes or it can assume that certain hierarchy in vegetation superiority exists. It seems in this case, the model assumes that shrubs are always superior to grasses and if they can exist then they will take over grasses. Is this correct? Is this a reasonable assumption. The purpose of additional description of the competition module is to highlight all primary assumptions and structure of the competition module while acknowledging its limitations. Yes, models aren't perfect but if their features and limitations are well highlighted then it's easier for readers to put the results in the context of the model.

Page8, Section 3.2, lines 15-16. " ... we subtract 100-year averages of consecutive time slices ..." only becomes clear once a reader looks at Figure 5. Please reword this sentence to make it more clear.

In Figure 5 the units of precipitation change make sense. The units of precipitation are mm/year and then the change is mm/year per 2k year. This can be simplified and referred to change in annual precipitation and then the units would just be mm/2k years. However, the units of change in fractional vegetation cover seem incorrect. What does fraction/year (i.e. year^(-1)) means? Why is there a year in the denominator? If change in fractional cover over 2k years is being referred to then units should just be fraction/2k years. I am unclear why there's an additional year^(-1) term needed.

Page 9, line 4. It took me a while to realize that delta_P and delta_veg_max do not refer to zonal averages but instead Figure 6 shows zonal averages of these quantities. Please consider rewording this sentence.

Page 9, lines 14-16. I wasn't able to follow this sentence.

Page 9, line 21. What is a "dominant branch"?

Page 10, towards the end of section 3.2, it is discussed how disappearance of SRG leads C4 grasses to establish and an increase in fractional vegetation cover for same precipitation. 1) Why does SRG disappears, and 2) isn't this behaviour (of higher fractional vegetation cover for same precipitation) unrealistic.

Page 10, line 9. " ... with a strong feedback between single plant types and climate". This sentence is unclear.

On page 10, and earlier on, does "realized PFTs" means the PFT that can potentially exist in a grid cell.

Page 10, last two sentences. Please explain "trait flexibility" and "evolutionary optimality hypothesis" in one or two sentences.

In context of issue raised by Reviewer #1 also consider showing absolute annual temperatures for 8k years ago and temperature change relative to 0k to justify the need for tropical tree PFT that can survive 10 degree Celsius coldest month temperature.

Figure 4 is an important figure. Figure 5 is also an important figure which illustrates whether the change in precipitation and fractional vegetation cover is gradual or immediate. However, overall as a reader I felt that this discussion wasn't enough or complete to convey the primary message around how the system operates. Perhaps, a simple cartoon of Figure 4 can be used to help understand a reader the discussion around Figure 4 e.g. using horizontal and vertical lines touching the Y and the X axes, respectively.

I am also attaching an annotated version of the manuscript with my hand written comments a lot of which I have already summarized here. But please see this version for other minor comments.

Please also note the supplement to this comment:
https://www.biogeosciences-discuss.net/bg-2017-425/bg-2017-425-RC2-supplement.pdf

[Figure]

**Supplement:**

[revised manuscript text omitted]

*How does use of present day SSTs affect your results.*

*→ these correspond to what time period/values*

**2.3 Modifications of plant functional diversity**

To investigate the effects of changes in plant functional diversity on climate–vegetation interaction towards the end of the AHP, we perform four different types of simulations: as a baseline serves the simulation $EXP_{ALL}$ with all natural PFTs commonly used in JSBACH (Tab. 1). The second type of simulation is a single-PFT experiment $EXP_{C4}$ which features only

C4 Grass ($C4$) and thereby excludes all woody PFTs and their competition in the study domain. Third, we combine the woody PFTs Tropical Evergreen Tree ($TE$) and Raingreen Shrub ($SRG$) with C4 Grass in the study domain for the experiments $EXP_{TE,C4}$ and $EXP_{SRG,C4}$. The *fourth*  type of simulation addresses the representation of bioclimatic limits in JSBACH. During the  present study, we found that the minimum temperatures of the coldest month fall in some regions in north Africa below the bioclimatic limit of Tropical Evergreen and Deciduous Trees (15.5°C) and thereby prevent their establishment in areas where reconstructions indicate the presence of tropical trees during the AHP (Hély et al., 2014). Although the definition of this limit is based on an empirical relationship between mean temperatures of the coldest month and absolute minimum temperature (frost occurrence) (Müller, 1982; Prentice et al., 1992), favorable microclimatic conditions could have allowed tropical taxa to establish. Another argument is that there might have been taxa that were partially frost tolerant and could survive short periods of freezing temperatures. A literature-based compilation of experimental cold tolerance thresholds for leaves (evergreens), buds and twigs or stem illustrates that "tropical evergreens show damage at - 1°C or -2°C. Many of the broad-leaved evergreens can tolerate - 10 to -15°C, with a few able to survive -20°C" (Harrison et al., 2010). In order to test the model sensitivity to this limit, we add a newly designed tropical frost tolerant woody PFT ($TD_{10}$) with relaxed bioclimatic limits to the standard PFT set in $EXP_{TD10}$. The minimum temperature of the coldest month is for $TD_{10}$ reduced from 15.5 to 10°C, *while* all other parameters  *remain same as for* the Tropical Deciduous Tree PFT ($TD$), see Tab. 1. In all simulations, the initial cover fractions are equally distributed over all included PFTs in the study domain.

In the following, we refer to the "potential" PFT diversity as the number of PFTs allowed in a simulation and to the "realized" PFT diversity as the number of PFTs that actually establish in a grid cell. We present the effects of changes in plant functional

*in the $EXP_{ALL}$ case.*

*How is $C_4$ albedo (which is grass)  lower than ALL which also includes darker trees?*

[revised manuscript text omitted]

---

## Author Comment (AC3) · 16 Feb 2018

**This manuscript aims to assess the sensitivity of the simulated extent of green Sahara during the African Humid Period (AHP), and the manner in which the land cover from this green Sahara transitions to current desert conditions, to the representation of plant functional types (PFTs) in the MPI-ESM (i.e. the PFT diversity). The subject of the manuscript is of broad interest and many past studies have addressed the question of the extent of green Sahara. Authors claim that differences in past studies are due to representation of different PFTs in different models (this appears to be a valid argument based on results presented in this manuscript) but the general strength of the land-atmosphere feedbacks (which depends on a particular land surface model) likely also plays a role. That is, the**

[Figure]

**extent of green Sahara will be different amongst models despite same PFTs because different land surface models have different strengths of land atmosphere coupling. In addition, different land surface models will likely represent same PFTs with different parameterizations and parameter values. This aspect is not discussed in the manuscript.**

The strength of land atmosphere coupling surely plays an crucial role for the extent of the green Sahara and has been considered in previous modelling studies, see references page 2, line 1. The new contribution to the question of the extent of the green Sahara in the present study is the explicit consideration of plant functional diversity – which has so far not been considered in models of higher complexity than the conceptual models by Claussen et al. (2013) and Groner et al. (2015) – at the example of JSBACH.

We do not claim that the representation of PFTs is the most and only important factor determining the extent of the green Sahara, though we illustrate that the choice of PFTs can have a significant impact on the climate-vegetation system. The representation of PFTs as well as the strength of land atmosphere coupling differ of course between models, but we expect that the observed effects are qualitatively not limited to JSBACH and might also apply to other models. A complete evaluation of effects on climate-vegetation interaction and the comparison between different land surface models goes beyond the scope of this study.

**This study can be assessed in the context of paleo vegetation as the first reviewer appears to have done but also in the context of land-atmosphere interactions which is where my expertise lies. In my opinion, the manuscript needs clarification of several items before it may be considered for publication in BG.**

**The competition between PFTs is an important aspect of this study but I felt that the description on page 4 (lines 5 – 25) wasn't sufficient for me as a reader to understand how competition actually works. Since this is a modelling study, I**

**feel it is important to lay it out for the reader. The text on page 5 attempts to do this but it seems it doesn't go all the way.**

As the model we used in this study was described in detail in previous publications, see eg. Brovkin et al. (2009), Reick et al. (2013), we confined ourselves to providing an brief overview over the most important aspects of competition in JSBACH. Based on your comment, we understand that further description is necessary here and we gladly adjust the manuscript to make it more comprehensible for the reader.

**Other comments**
**Having read the authors' response to Reviewer #1 I now realize out of 21 PFTs only 8 PFTs can grow in the region considered. However, when I read the manuscript the first time I had similar confusion. So perhaps this point needs to be clarified.**

We understand that the phrasing is confusing here and we gladly adjust the manuscript to make it more clear.

**Section 2.2 needs more info about model setup and discussion of implications of how this set up is done.**
**1) How does use of present day SSTs affects the overall results? Does an interactive ocean generally amplify or dampen the effect of land-atmosphere feedbacks?**

It has been shown that the west African monsoon and the seasonal cooling of the equatorial Atlantic amplify each other, see e.g. Okumura et al. (2004). Thus, the use of present day SST probably reduces the overall strength of the west African monsoon in mid-Holocene simulations. Thereby the simulated precipitation and vegetation cover fractions are probably lower than with AHP SST. We would expect an interactive ocean to generally amplify the monsoon signals caused by changes in PFT composition.

Our study focuses on the effect of plant functional diversity rather than synergies that

might occur with an interactive ocean. As we prescribed SST identically for all simulations, we do not expect qualitative differences in the results. Nevertheless, we gladly add this information to the manuscript.

**2) Other atmospheric boundary conditions correspond to what time period?**

Other atmospheric boundary conditions are prescribed after IPCC AR5 (2013). Tropspheric methane is by default prescribed with a constant mixing ration of 1.69 ppmv. N2O is by default prescribed with a constant mixing ration of 309 ppbv. CFC11 and CFC12 are prescribed with constant mixing ratios of 0.253 ppbv and 0.466 ppbv, respectively. Other CFC species are currently not included. The default ozone climatology is given as three-dimensional monthly values, although in the stratosphere, ozone is assumed to not vary with longitude, and is based on the merged and future ozone climatology as described by Cionni et al. (2011) for CMIP5. The stratospheric aerosol is based on an extension of the Pinatubo aerosol data set (Stenchikov et al., 1998) to cover the entire period between 1850 and 1999. We will add this information to the manuscript.

**3) What do soil properties mean – do you mean soil texture and permeable soil depth.**

By soil properties we refer here mainly to the soil depth until the bedrock and the water holding capacity. These properties are generated from a compilation of data sets, provided as input maps to the model (Hagemann et al., 2014). Further parameters and their references are summarized in Hagemann et al. (2014). For our study it is most relevant that in the 5-layer scheme soil scheme in JSBACH (Hagemann et al., 2014) the number of active layers is limited by the depth until the bed rock. Thus, the soil water content may be greater than 0 only for those layers with a soil depth above the bedrock as there is no water available for the land surface scheme within the bedrock. Soils in the present day Sahara are very shallow compared to the mid-Holocene, especially in mountainous regions, and have a very low water holding capacity making growth

difficult for PFTs with high moisture requirements even though provided precipitation is sufficient.

Accounting for the changes in soil properties during the AHP would probably further amplify the feedback between land surface and atmosphere. However, as mentioned above, we focus here on the effects of plant functional diversity. Nevertheless, we gladly add more information about soil properties to the manuscript.

**4) Since land-atmosphere feedbacks are key to understanding the results presented in this manuscript it would be useful to put albedo, typical LAI, rooting depth and vegetation height of different PFTs in a table for reader to understand how the different physical characteristics of a PFT can potentially affect land-atmosphere interactions.**

We gladly extend the table by albedo values for the considered PFTs. The maximum LAIs listed in Tab. 1 can be assumed as typical LAIs for the presented PFTs since the LAI quickly increases to that value in an asymptotic way as soon as growth conditions are favourable in terms of moisture availability, bioclimatic limits and growing season. The rooting depth is assumed to be the same for all PFTs following the rooting depth map provided by Hagemann et al. (2014). We will add a description to manuscript. JSBACH has no explicit implementation of vegetation height, it is implicitly considered through the roughness length, a parameter we can easily attach to the table for clarification.

**On Page 6, around line 24, it is mentioned albedo in the C4 grass simulation is lower than that in the $EXP_{ALL}$ simulation. Is this an error? Grasses are generally brighter than trees. But then on page 7, lines 11 and 12, the manuscript correctly notes that albedo is higher in $EXP_{C4}$ than in $EXP_{ALL}$. Please correct the sentences on page 6 so that they are consistent with the text on page 7.**

The manuscript is correct here, we consider different regions in the two examples you mentioned. In the first example, we describe the effects in region 1, where grasses are

darker compared to the bare soil that prevails in $EXP_{ALL}$. The second example refers to region 2 where grasses are brighter compared to the trees that prevail in $EXP_{ALL}$. We agree that the formulation is not clear and we gladly adjust the manuscript to highlight the difference.

**On Page 7, line 2, the phrase "slows down water and energy fluxes" is unclear. What does slowing down means does it mean decrease in fluxes. If yes, which fluxes – I suppose evapotranspiration (i.e. latent heat). Please be explicit.**

By slowing down we mean here that the absolute fluxes decrease, including sensible and latent heat flux, evapotranspiration and precipitation. We will adjust the formulation for clarification.

**Page 7, line 16-20. Yes, C4 grasses are more productive than C3 grasses. But productivity is different than specific leaf area (SLA, m2/kg C) which is a measure of how many m2 of leaves can be constructed with a Kg of carbon of leaf biomass. The discussion in lines 16-20 appears to be mixing productivity with SLA.**

We agree that the formulation is not explicit here, we will adjust the formulation for clarification.

**Page 7, line 21. "Grass cover is not directly reduces by disturbances". This seems contradictory to what happens in nature. Grasses are more flammable than trees so fires affect grass cover more drastically – although, of course, grasses spring back faster too.**

We understand your confusion here and we agree with your argumentation. Grasses are of course affected by fire, they burn in JSBACH just like other PFTs. What we wanted to express here is that the cover fraction of grasses does not change if fire occurs because grasses automatically establish on uncolonized land in the following year. We will adjust the formulation for clarification.

**Page 7, line 33. "...SRG outcompetes C4 due to the implicit assumption of light competition". So are shrubs assumed to be taller than trees. This is where a more complete description of how competition works can help. A model can simulate the actual physical processes or it can assume that certain hierarchy in vegetation superiority exists. It seems in this case, the model assumes that shrubs are always superior to grasses and if they can exist then they will take over grasses. Is this correct? Is this a reasonable assumption.**

We agree that a more detailed description of the competition is necessary to follow our argumentation here. The model assumes a hierarchy in vegetation superiority. Shrubs belong to the class of woody PFTs and are therefore always superior to grasses, among other things because they are assumed to be taller, see roughness length. We assume this to be an ecologically reasonable assumption.

**The purpose of additional description of the competition module is to highlight all primary assumptions and structure of the competition module while acknowledging its limitations. Yes, models aren't perfect but if their features and limitations are well highlighted then it's easier for readers to put the results in the context of the model.**

We agree that mentioning the limitations of a model is just as important as the description of its features, thus we gladly include a brief summary in the manuscript. In terms of vegetation dynamics, one important limitation is surely the representation of plant diversity with a static set of discrete PFT parameters which does not cover the range of species categorized as one PFT and disregards phenotypic plasticity and trait variability which are often larger within PFTs than between PFTs (see page 10, line 27-29). Additionally, the fauna which plays an important role in savannas in terms of grazing is not represented in JSBACH, and nutrient availability which is a limiting factor of plant growth is not considered.

**Page8, Section 3.2, lines 15-16. "...we subtract 100-year averages of consecutive**

**time slices..." only becomes clear once a reader looks at Figure 5. Please reword this sentence to make it more clear.**

We will reword this sentence for clarification as follows "For a first estimate of regional transition patterns from the "green" Sahara (8 ky) to the "desert" state (0 ky), we calculate differences between consecutive time slices (8 ky-6 ky, 6 ky-4 ky, 4 ky-2 ky, 2 ky-0 ky) of the 100-year averages of $P$ and $veg_{max}$ for all simulations, and compare the resulting transition maps of $P$ and $veg_{max}$ in the whole study domain (12 to 34° N, -15 to 40° E).".

**In Figure 5 the units of precipitation change make sense. The units of precipitation are mm/year and then the change is mm/year per 2k year. This can be simplified and referred to change in annual precipitation and then the units would just be mm/2k years. However, the units of change in fractional vegetation cover seem incorrect. What does fraction/year (i.e. yearËĘ(-1)) means? Why is there a year in the denominator? If change in fractional cover over 2k years is being referred to then units should just be fraction/2k years. I am unclear why there's an additional yearËĘ(-1) term needed.**

We choose these units because the change in each 100-year averaged value is mm/yr for precipitation and fraction/yr for vegetation cover fraction and we want to represent the difference between two consecutive time slices, thus in 2000 years. The change is not mm/2000yr but mm/yr over 2000 years. The same accounts for vegetation cover fraction with the change in fraction/yr in 2000 years.

**Page 9, line 4. It took me a while to realize that $delta_P$ and $delta_{veg_{max}}$ do not refer to zonal averages but instead Figure 6 shows zonal averages of these quantities. Please consider rewording this sentence.**

We will reword this sentence for clarification as follows "For the quantitative comparison of all simulations, we condense the information of the transition maps by calculating zonal means (-15 to 40° E) of $P$ and $veg_{max}$ and subtracting consecutive time

slices. These zonally averaged differences between consecutive time slices are in the following referred to as $\Delta P$ and $\Delta veg_{max}$."

**Page 9, lines 14-16. I wasn't able to follow this sentence.**

With this sentence we want to describe that the initial values of vegetation cover fraction and precipitation contribute to the potential rate of decline. If the initial values are low, there is no potential to show a decrease a large as starting from a higher value. We will rephrase this part for clarification.

**Page 9, line 21. What is a "dominant branch"?**

The dominant branch refers to the curve most of the points lie on in contrast to the outlying data points. We understand that this formulation is confusion, we will rephrase the sentences including that formulation.

**Page 10, towards the end of section 3.2, it is discussed how disappearance of SRG leads C4 grasses to establish and an increase in fractional vegetation cover for same precipitation. 1) Why does SRG disappears, and 2) isn't this behaviour (of higher fractional vegetation cover for same precipitation) unrealistic.**

SRG disappears because the environmental conditions are no longer supporting its growth, in particular precipitation is too low. The increase in vegetation cover fraction of C4 can be explained by the lack of competition from SRG. All available water can be used by C4 and as it has a lower SLA, it is able to cover a larger area with less precipitation. That is an ecologically reasonable behavior.

**Page 10, line 9. "...with a strong feedback between single plant types and climate". This sentence is unclear.**

We will reword this sentence as follows "Our findings thereby reconcile a gradual transition from a "green" state to a "desert" state with a strong feedback between vegetation and climate."

**On page 10, and earlier on, does "realized PFTs" means the PFT that can potentially exist in a grid cell.**

As described on page 5, line 32-33, we refer to the "potential" PFT diversity as the number of PFTs allowed in a simulation and to the "realized" PFT diversity as the number of PFTs that actually establish in a grid cell.

**Page 10, last two sentences. Please explain "trait flexibility" and "evolutionary optimality hypothesis" in one or two sentences.**

We will reword this part as follows "Alternative approaches to represent plant diversity consider the simulation of individual plants (e.g. LPJ-GUESS (Smith et al., 2001); aDGVM (Scheiter et al., 2013), "trail variability" which allows selected traits to vary within the range of observations to optimize growth under the environmental conditions (e.g. JSBACH (Verheijen et al., 2013,2015), "trait flexibility" which represents plant diversity in terms of plant ecophysiological trade-offs instead of PFTs by selecting for the most suitable growth strategy out of randomly generated sets of parameter values (e.g. JeDi-DGVM (Pavlick, 2012); aDGVM Scheiter et al., 2013)), or operate based on the "evolutionary optimality hypothesis", an approach based on the microeconomic standard framework to determine the optimal input mix for a two-input production process, here water loss and carbon gain during photosynthesis (Wang et al., 2017). "

**In context of issue raised by Reviewer #1 also consider showing absolute annual temperatures for 8k years ago and temperature change relative to 0k to justify the need for tropical tree PFT that can survive 10 degree Celsius coldest month temperature.**

Although winter temperatures were on average lower at 8 ky than at 0 ky, the affected region (around 18 to 22 deg N) is slightly warmer at 8 ky in our simulations. However, the minimum temperature of the coldest month for tropical trees falls below the threshold of 15.5 deg Celsius at 8k as well as 0 ky. The difference is that at 8 ky, precipitation is substantially higher than at 0 ky which allows tropical trees to sustain

growth as soon as established. Therefore, we did not to include a figure of temperature differences in the manuscript, but we agree that an additional explanation in Sec. 2.3 will serve clarification.

**Figure 4 is an important figure. Figure 5 is also an important figure which illustrates whether the change in precipitation and fractional vegetation cover is gradual or immediate. However, overall as a reader I felt that this discussion wasn't enough or complete to convey the primary message around how the system operates. Perhaps, a simple cartoon of Figure 4 can be used to help understand a reader the discussion around Figure 4 e.g. using horizontal and vertical lines touching the Y and the X axes, respectively.**

We understand that the interpretation of Fig. 4 might be difficult without further illustration of the thresholds we discuss in the text. We gladly include supporting lines in the graphs to make it easier to capture the differences between the experiments.

**I am also attaching an annotated version of the manuscript with my hand written comments a lot of which I have already summarized here. But please see this version for other minor comments.**

We will consider your comments in the revision of the manuscript, thank you very much for the careful evaluations.